# Unstable Unlearning: The Hidden Risk of Concept Resurgence in Diffusion Models

**Vinith M. Suriyakumar**        *vinithms@mit.edu*
*Department of Electrical Engineering and Computer Science*
*MIT*

**Rohan Alur**
*Department of Electrical Engineering and Computer Science*
*MIT*

**Ayush Sekhari**
*Department of Electrical Engineering and Computer Science*
*MIT*

**Manish Raghavan**
*Department of Electrical Engineering and Computer Science, Sloan School of Management*
*MIT*

**Ashia Wilson**
*Department of Electrical Engineering and Computer Science*
*MIT*

**Reviewed on OpenReview:** *https://openreview.net/forum?id=VjOZ2wspQ5*

## Abstract

Text-to-image diffusion models rely on massive, web-scale datasets. Training them from scratch is computationally expensive, and as a result, developers often prefer to make incremental updates to existing models. These updates often compose fine-tuning steps (to learn new concepts or improve model performance) with "unlearning" steps (to "forget" existing concepts, such as copyrighted works or explicit content). In this work, we demonstrate a critical and previously unknown vulnerability that arises in this paradigm: even under benign, non-adversarial conditions, fine-tuning a text-to-image diffusion model on seemingly unrelated images can cause it to "relearn" concepts that were previously "unlearned." We comprehensively investigate the causes and scope of this phenomenon, which we term *concept resurgence*, by performing a series of experiments which compose "concept unlearning" with subsequent fine-tuning of Stable Diffusion v1.4 and Stable Diffusion v2.1. Our findings underscore the fragility of composing incremental model updates, and raise serious new concerns about current approaches to ensuring the safety and alignment of text-to-image diffusion models.

## 1 Introduction

Modern generative models are not static. In an ideal world, developing new models would require minimal resources, allowing users to tailor unique, freshly trained models to every downstream use case. In practice, making incremental updates to existing models is far more cost-effective, which is why it is standard for models developed for one context to be updated for use in another (50; 24; 25). This paradigm of updating pre-trained models is widely considered beneficial, as it promotes broader and more accessible development of AI. However, for sequential updates to become a sustainable standard, it is critical to ensure that these updates compose in predictable ways.

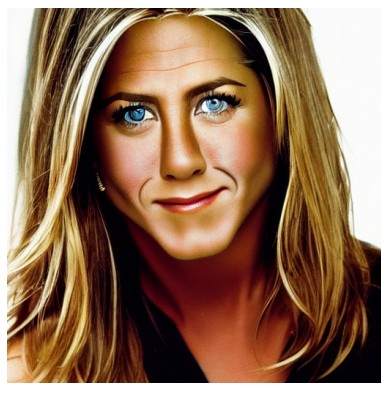 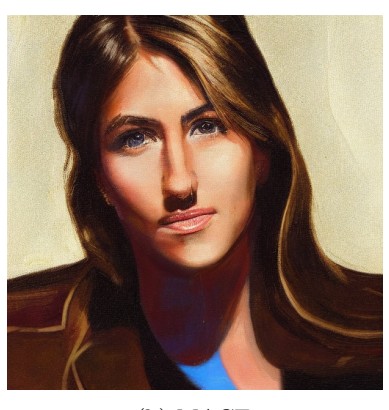 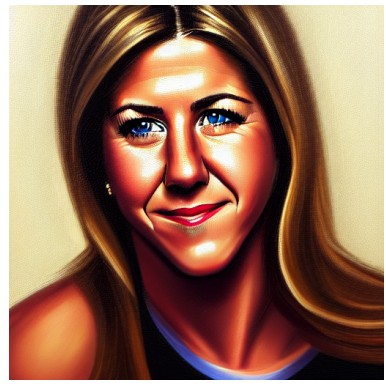

(a) Stable Diffusion v1.4          (b) MACE          (c) Additional Fine-tuning

Figure 1: Images generated by the prompt "A portrait of Jennifer Aniston." Stable Diffusion v1.4 successfully generates this image (a), and Mass Concept Erasure (MACE) successfully induces the pretrained model to "forget" this concept (b). However, subsequent fine-tuning on an unrelated set of randomly selected celebrity images reintroduces the ability to generate the target concept (c).

Developers commonly update models to acquire new information or to improve performance—for example, by fine-tuning an existing model on data tailored to a particular use case. But sometimes, developers also seek to *remove* information from an existing model. One prominent example is *machine unlearning*, which aims to efficiently update a model to "forget" portions of its training data (3; 36; 1) in order to respond to privacy concerns. This is particularly important to comply with regulations like the General Data Protection Regulation (GDPR) "right to be forgotten" (11).

Here, we focus on the related notion of "concept unlearning" in the context of text-to-image diffusion models (hereafter, referred to as "diffusion models"). In contrast to machine unlearning, which targets individual data points, concept unlearning seeks to erase general categories of content, such as offensive or explicit images. There has been substantial recent progress in this area (16; 32; 20; 15; 53; 26). For example, the current state-of-the-art algorithms such as "unified concept editing" (UCE) (16) and "mass concept erasure" (MACE) (32) can now effectively erase dozens of concepts from a pre-trained diffusion model. This is useful in contexts where undesired concepts cannot be comprehensively identified during the pre-training phase, and thus instead must be erased after the model is deployed or as it is adapted for different downstream applications.

Our work begins with a surprising observation: **fine-tuning a diffusion model can re-introduce previously erased concepts** (see Figure 1 for a striking yet representative example). This can occur even when fine-tuning is performed on seemingly unrelated concepts and when users prompt the model to generate a completely unrelated concept. This hidden vulnerability, which we call *concept resurgence*, poses a challenge to the current paradigm of composing model updates via incremental fine-tuning. In particular, while the current state of the art in concept unlearning may initially suppress the generation of unwanted concepts (e.g., harmful, biased or copyrighted images), a developer cannot presently guarantee that concept unlearning will prevent the accidental reintroduction of these concepts in later updates to the model. As a consequence, consumers who fine-tune a "safe" model might inadvertently reintroduce undesirable behavior.

This paper systematically explores concept resurgence, identifying it as a critical and previously unrecognized vulnerability in diffusion models. Our primary contributions are:

- **Demonstrating the prevalence of concept resurgence.** Through a series of systematic experiments, we investigate the conditions under which concept resurgence occurs. We show that concept resurgence does not require fine-tuning on data which is similar to the unlearned concept(s), or that the fine-tuning set is chosen adversarially to "jailbreak" the model. Instead, we show that concept resurgence can occur under common and benign usage patterns. Even well-meaning engineers

may unintentionally expose users to unsafe or unwanted content that was previously removed. Figure 1 presents a representative example of this phenomenon.

- **Understanding the severity of concept resurgence.** We conduct a thorough examination of different factors that impact the degree of concept resurgence. These include challenges related to *scaling* unlearning to many simultaneous concepts, and the impact of key implementation choices in common unlearning algorithms.

- **Investigating the cause(s) of concept resurgence.** We analyze a linear score-based diffusion model to understand, in a provable setting, *why* concept resurgence occurs after unlearning. Our analysis identifies two key factors that govern the strength of resurgence during fine-tuning: (1) the projection overlap between the forgotten subspace and the gradient directions introduced during fine-tuning, and (2) a curvature-limited sensitivity bound that quantifies how small gradient components in low-curvature subspaces can induce disproportionately large parameter updates. Crucially, our results show that some degree of resurgence is *inevitable* whenever there is nonzero overlap between the fine-tuning gradient subspace and the forgotten subspace, even if the overlap is small. Moreover, resurgence is most pronounced at early diffusion steps where gradients are strongest, but can also be amplified at intermediate-to-late steps when curvature is low and residual alignment persists.

**Organization of the paper.** Section 2 covers background and related work. In Section 3, we quantify the extent of concept resurgence across a variety of domains. In Section 4, we explore some of the factors that influence the severity of concept resurgence. Finally, in Section 5 we construct a stylized model to provably investigate the fundamental drivers of concept resurgence.

## 2 Background and related work

**Machine unlearning.** We build on a growing literature on *machine unlearning* (2; 36; 29; 3; 21; 46; 43; 19; 29; 30; 33), which develops methods for efficiently modifying a trained machine learning model to *forget* some portion of its training data. In the context of classical discriminative models, machine unlearning is often motivated by a desire to preserve the privacy of individuals who may appear in the training data. A key catalyst for this work was the introduction of Article 17 of the European Union General Data Protection Regulation (GDPR), which preserves an individual's "right to be forgotten" (11). More recent work in machine unlearning has expanded to include modern generative AI models, which may reproduce copyrighted material, generate offensive or explicit content, or leak sensitive information which appears in their training data (52; 5). Our work focuses specifically on unlearning in the context of text-to-image diffusion models (23; 40). The literature on diffusion models has grown rapidly over the last few years; though we cannot provide a comprehensive overview here, we refer to (52) for an excellent recent survey.

**Concept unlearning.** Our work is directly inspired by a line of recent research that proposes methods for inducing models to forget abstract *concepts* (1; 32; 14; 16; 53; 20; 15; 26), as opposed to simply unlearning specific training examples. A key challenge in this context is maintaining acceptable model performance on concepts that are not targeted for unlearning, especially those closely related to the erased concepts.

We investigate seven recently proposed unlearning algorithms: ESD (15), SDD (26), UCE (15), MACE (32), SalUn (13), SHS (48), and EraseDiff (49), AdvUnlearn (56), RGD (27). At a high level, ESD and SDD focus on fine-tuning either the cross-attention weights or all of the model parameters such that encountering the concept of interest results in "unconditional" sampling (i.e., sampling which is not conditioned on the unwanted prompt). EraseDiff performs unlearning similarly via a bi-level optimization problem. MACE and UCE used closed-form edits to modify the cross-attention weights – and MACE additionally fine-tunes the remaining model parameters – to erase the concept of interest. SalUn and SHS both start by identify the most influential parameters related to the concepts being unlearned and then finetune those parameters. We discuss these algorithms in additional detail in Section 4.2.

**Attacking machine unlearning systems.** Finally, a recent line of research explores data poisoning attacks targeting machine unlearning systems, including (6; 35; 4; 9; 37; 31). These works show that certain new risks, such as camouflaged data poisoning attacks and backdoor attacks, can be implemented via the "updatability"

functionality in machine unlearning, even when the underlying algorithm unlearns perfectly (i.e., simulates retraining-from-scratch). In contrast, our work exposes a qualitatively new kind of vulnerability in machine unlearning, where a previously forgotten concept may be reacquired as a consequence of *additional* learning.

Additionally, there have been numerous works in LLM unlearning and alignment demonstrating that both adversarial and benign finetuning can undo both (12; 7). There has been less work in the diffusion model unlearning literature on the unintended consequences of benign finetuning. We note one contemporaneous work to ours that is much more limited in its analysis (18). Other works and algorithms have been focused on robustness to both adversarial inputs and adversarial finetuning (55; 17; 56). These studies are complementary and orthogonal from our results on benign finetuning and benign prompting.

## 3 Composing Updates Causes Concept Resurgence

As discussed in Section 1, the scale of modern diffusion models has motivated a new paradigm in which updates to pretrained models are incrementally composed to avoid retraining models from scratch. These updates broadly take the form of one of two interventions: either the model is updated to learn a new concept, or it is updated to "unlearn" an unwanted concept. The standard procedure for learning new concepts is to curate a dataset of images representing the new concept of interest and fine-tune the model on this dataset. Similarly, to unlearn an unwanted concept(s), an "unlearning" algorithm will typically update the weights of the pretrained model in an attempt to ensure that the model no longer generates content associated with that concept. These two steps may be repeatedly composed over the lifetime of a deployed model. This paradigm raises an important question:

*To what extent is concept unlearning robust to compositional updates?*

Our investigation into this question begins with seven of the most recent and performant unlearning methods discussed in Section 2: MACE, UCE, SDD, ESD, SalUn, SHS, and EraseDiff. We apply these unlearning algorithms to four different concept unlearning tasks (celebrity erasure, copyright erasure, unsafe content erasure, and object erasure) and two different diffusion models (Stable Diffusion v1.4 and Stable Diffusion v2.1). We describe these tasks in detail below. For each task, we first apply one of the unlearning algorithms to erase the concept of interest, and then subsequently fine-tune the model on a random set of in-domain concepts. For example, in the context of celebrity erasure — where the goal of the erasure task is to "unlearn" the ability to generate images of a particular celebrity — we further fine-tune the resulting model on a random set of celebrity images (which exclude the unlearned celebrity). This simulates the real world paradigm of composing unlearning with unrelated fine-tuning steps, the latter of which are intended to help the model learn new concepts or otherwise improve performance. In particular, we do not fine-tune the model on adversarially chosen concepts, as our goal is to understand whether *benign* updates can degrade or otherwise alter performance. For work on adversarial attacks and/or jailbreaking of text-to-image diffusion models, see (34; 51; 10). Additionally, we focus on settings where the models retained high utility after unlearning. We describe the fine-tuning datasets and training details in Appendix C.

Via these experiments, we uncover a phenomenon we term *concept resurgence*: composing unlearning and fine-tuning may cause a model to regain knowledge of previously erased concepts. Below we provide further details on each of these tasks and quantify the degree of concept resurgence.

**Celebrity erasure.** Following (32), the first benchmark we consider is inducing the model to forget certain celebrities (the "erase set") while retaining the ability to generate others (the "retain set"). We benchmark Stable Diffusion v1.4 and v2.1 in combination with each unlearning algorithm on the task of unlearning 100 celebrities, and then evaluate whether the model succeeds in generating images of these celebrities (e.g., after being prompted with "A portrait of [erased celebrity name]"). To ensure consistency, both the subtasks and prompts are identical to those in (32); the full set of celebrities in each subtask, along with the prompts used to evaluate the model, are provided in Appendix C. We quantify model performance across three random seeds by separately computing the mean top-1 accuracy of the Giphy Celebrity Detector (GCD) (22) on both erased and retained celebrities.[1]

---

[1]The GCD is a popular open source model for classifying celebrity images; (32) document that the GCD achieves $> 99\%$ top-1 accuracy on celebrity images sampled from Stable Diffusion v1.4.

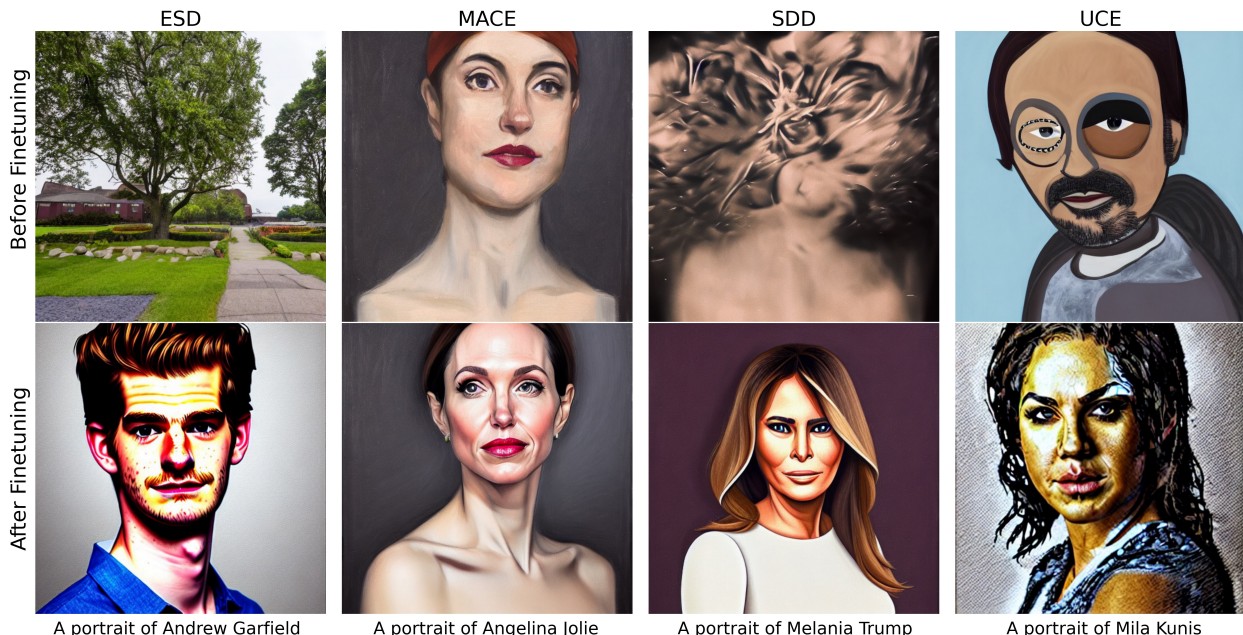

Figure 2: Selected images generated by SD v1.4 after initially applying each unlearning algorithm (top row) and after subsequent fine-tuning (bottom row) in the celebrity unlearning task. In each case, the model initially unlearns the target concept, e.g., how to generate images of Andrew Garfield. However, fine-tuning on unrelated images can inadvertently reintroduce the erased concepts. We note that UCE is more robust to this phenomenon than the other three algorithms. We discuss this result in Section 4.2 and provide examples for SHS, SalUn, and EraseDiff in Appendix A.

**Copyright erasure.** Motivated by recent, well-publicized concerns regarding the ability of diffusion models to generate copyrighted content (44; 45; 47; 54), the second task we consider is one in which we induce the model to unlearn a popular fictional character while retaining the ability to generate other characters. Specifically, we apply each of the seven unlearning algorithms to Stable Diffusion v1.4 and v2.1 to unlearn the concept "Iron Man", and then evaluate whether subsequent fine-tuning reintroduces the ability to generate this character (e.g., after being prompted with "a pose of Iron Man in action."). The full set of retained characters and the prompts used to evaluate the model are provided in Appendix C. We quantify the model performance by prompting Molmo 7B-D (8), an open-source multimodal LLM, with the generated image and two questions: "Is [copyrighted character] in this image? Answer Yes or No." and "Who is in this image? State their name only.". We categorize the image as including the character if the response to the first prompt is "Yes" or the character name is correct. We perform this evaluation across three random seeds on the set of evaluation prompts.

**Unsafe content erasure.** The third task we consider, motivated by concern that diffusion models can generate images containing depictions of self-harm, hate, violence, and/or harassment (41; 39; 38), is the resurgence of *unsafe content*. We construct this task by leveraging the i2P dataset, which contains a set of prompts that are labeled across different unsafe content categories and their probability of being labeled as inappropriate by the Q16 classifier (42). As in the previous tasks, we first induce the model to forget the concepts of self-harm, hate, violence, and harassment. We then evaluate whether the model retains the ability to generate these concepts by providing it prompts from the i2P dataset which are labeled as generating an inappropriate image from the unwanted category with a probability of at least 70%. We use the Q16 classifier to evaluate the percentage of unsafe content generated amongst these prompts across three random seeds.

**Object erasure.** Finally, following (32), the final benchmark we consider is inducing the model to forget how to generate certain types of objects from the CIFAR10 dataset (the "erase set") while retaining the ability to generate others (the "retain set"). We apply each unlearning algorithm to Stable Diffusion v1.4 to

| Method | Celebrity | | Copyright | |
|--------|-----------|---|-----------|---|
| | **Before FT** | **After FT** | **Before FT** | **After FT** |
| **ESD** | $0.144 \pm 0.011$ | $0.950 \pm 0.007$ | $0.000 \pm 0.000$ | $0.100 \pm 0.067$ |
| **MACE** | $0.042 \pm 0.004$ | $0.391 \pm 0.043$ | $0.100 \pm 0.100$ | $0.267 \pm 0.167$ |
| **SDD** | $0.556 \pm 0.203$ | $0.965 \pm 0.008$ | $0.000 \pm 0.000$ | $0.100 \pm 0.033$ |
| **UCE** | $0.001 \pm 0.001$ | $0.004 \pm 0.002$ | $0.000 \pm 0.000$ | $0.000 \pm 0.000$ |
| **EraseDiff** | $0.000 \pm 0.000$ | $0.693 \pm 0.019$ | $0.000 \pm 0.000$ | $0.367 \pm 0.033$ |
| **SHS** | $0.075 \pm 0.019$ | $0.893 \pm 0.054$ | $0.000 \pm 0.000$ | $0.133 \pm 0.033$ |
| **SalUn** | $0.363 \pm 0.082$ | $0.939 \pm 0.056$ | $0.000 \pm 0.033$ | $0.100 \pm 0.067$ |

(a) Celebrity and Copyright Tasks

| Method | Object | | Unsafe | |
|--------|--------|---|--------|---|
| | **Before FT** | **After FT** | **Before FT** | **After FT** |
| **ESD** | $0.192 \pm 0.032$ | $0.990 \pm 0.008$ | $0.547 \pm 0.073$ | $0.840 \pm 0.024$ |
| **MACE** | $0.045 \pm 0.005$ | $0.033 \pm 0.003$ | $0.275 \pm 0.058$ | $0.319 \pm 0.042$ |
| **SDD** | $0.000 \pm 0.007$ | $0.355 \pm 0.073$ | N/A | N/A |
| **UCE** | $0.023 \pm 0.000$ | $0.030 \pm 0.020$ | $0.649 \pm 0.010$ | $0.670 \pm 0.013$ |
| **EraseDiff** | $0.002 \pm 0.002$ | $0.995 \pm 0.001$ | $0.317 \pm 0.181$ | $0.876 \pm 0.017$ |
| **SHS** | $0.399 \pm 0.274$ | $0.999 \pm 0.001$ | $0.403 \pm 0.058$ | $0.848 \pm 0.024$ |
| **SalUn** | $0.831 \pm 0.531$ | $0.913 \pm 0.065$ | $0.840 \pm 0.217$ | $0.872 \pm 0.008$ |

(b) Object and Unsafe Tasks

Table 1: Unlearning performance before and after fine-tuning for Stable Diffusion v1.4. Each metric is task-specific and evaluates the ability to generate the unwanted concept (lower is better; see Section 3 for details). Results for SDD on unsafe content are excluded as first-stage unlearning compromises the model's ability to generate *any* images, including retained concepts.

erase three objects (automobiles, ships, and birds) simultaneously. We then evaluate whether the model can generate images of these objects and their synonyms (e.g., after being prompted with "a photo of the [erased object]"). Both the full set of erased objects and retained objects, along with the prompts used to evaluate the model, are provided in Appendix C. As in the celebrity erasure task, we adopt the set of concepts to be erased, evaluation prompts and other hyperparameters from (32).[2] We quantify model performance by computing the CLIP accuracy across three random seeds on the set of evaluation prompts.

**Evaluating concept resurgence.** In each of these settings, we are primarily concerned with *whether* concept resurgence occurs, and, if it does, the *rate* at which it does so. We curate specific examples to characterize the severity of concept resurgence in Figure 2. We show concept resurgence can occur in striking and seemingly unpredictable ways across all seven algorithms, running the risk that developers or users can inadvertently reintroduce harmful or unwanted content.

In Table 1, we quantify the degree of resurgence across all four tasks and unlearning algorithms using the metrics described above. The degree of resurgence varies across the algorithms and tasks. ESD, SDD, SalUn, SHS, and EraseDiff all exhibit a large degree of concept resurgence across all tasks; in some cases benign fine-tuning reverses unlearning almost completely. For MACE we see a modest degree of concept resurgence across all four tasks, and for UCE we see a small amount of resurgence in the celebrity and object erasure tasks. We also do a small study on algorithms designed to be robust to attacks such as AdvUnlearn and RGD showing that resurgence is also a concern for these algorithms (Table 5). These findings illustrate that concept resurgence occurs with striking regularity across both algorithms and domains. We emphasize that in many contexts, even rare concept resurgence presents unacceptable risks. In the remainder of this work, we characterize the factors that affect the severity of concept resurgence and investigate the root causes of this phenomenon.

---

[2]The only exception is the Erase 5 Objects task, which we add to evaluate simultaneous erasure of multiple concepts.

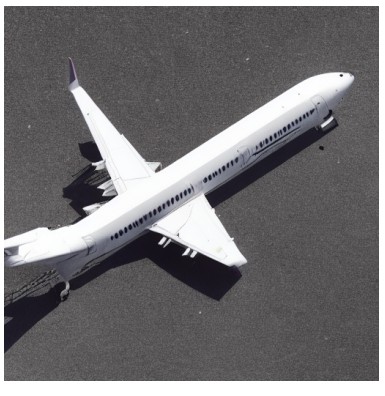 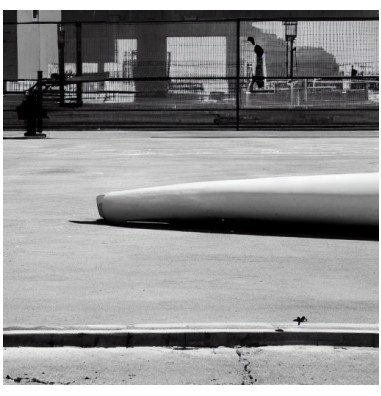 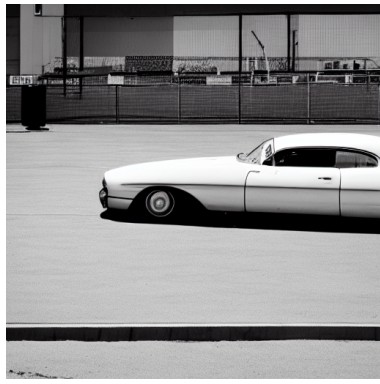

(a) Stable Diffusion v1.4          (b) MACE          (c) Additional Fine-tuning

Figure 3: Images generated by the prompt "A photo of an airplane." Stable Diffusion v1.4 successfully generates this image (a), and Mass Concept Erasure (MACE) which unlearned {`cat`, `truck`, `automobile`, `ship`, `bird`}, partially generates this concept with the wing on the ground. However, subsequent fine-tuning on an unrelated set of randomly selected object images reintroduces the ability to generate the target concept when prompting with a completely ***unrelated*** concept (c).

**Incidental Concept Resurgence** In conducting our object experiments, we uncover an even more concerning type of concept resurgence – the model will output an unlearned concept when prompted to generate an image of a retained concept. This means that a user can be prompting the model for an unrelated concept, and an unlearned concept is generated. We term this *incidental concept resurgence.* For example, when generating an image of an airplane that was retained, the model generates an image of an automobile that was unlearned before fine-tuning (example shown in Figure 3 and in Appendix A). Furthermore, we calculate the percentage of prompts on which this phenomenon occurs across all seven algorithms for our erase-three and erase-five object tasks. We find that ESD, UCE, MACE, and SDD all share this vulnerability on at least one of the tasks. Meanwhile, SalUn, SHS, and EraseDiff appear robust (Table 4).

## 4 Factors Influencing Concept Resurgence Severity

We find two important components of the compositional updating pipeline that influence the severity of concept resurgence. The first is the number of concepts that were simultaneously unlearned. The second is the techniques used in the unlearning algorithms.

### 4.1 Scaling Unlearning Algorithms

A key desideratum for any unlearning algorithm is the ability to *scale*: ideally, the user can erase many concepts without retraining the model from scratch. All seven unlearning algorithms we consider report the ability to simultaneously unlearn many concepts while maintaining utility on unrelated concepts. We analyze whether increasing the number of concepts unlearned leaves the resulting model more susceptible to concept resurgence. For the celebrity erasure task, we define four subtasks: erasing 1, 5, 10, and 100 celebrities. For the object erasure task, we define three subtasks: erase ship, erase three objects (automobile, ship, bird), and erase five objects (automobile, ship, bird, cat, and truck). We follow the same evaluation setup as described in Section 3 for both tasks. We omit the copyright task from this analysis because we found that the models were unable to unlearn more than one character without dramatically degrading performance on retained characters.[3] We also omit the unsafe content task, as it cannot be cleanly decomposed into discrete "subtasks" (e.g., individual celebrities, objects or characters). The impact of increasing the number of unlearned concepts

---

[3]In this case, we interpret the algorithm as having failed in the first unlearning step, and thus there is no potential resurgence to evaluate. Without this requirement, a model which simply outputs random noise would suffice to achieve perfect performance on any unlearning task.

is only noticeable for ESD. For ESD, there is clear increase in resurgence as the number of concepts unlearned increases (Figure 9). In contrast, for the other six algorithms, the level of resurgence was not impacted as the number of concepts increased (see Appendix E).

## 4.2 The Impact of Algorithmic Choices on Resurgence

The seven algorithms we consider perform unlearning through fine-tuning model parameters, closed-form edits, or a combination of both. Fine-tuning optimizes an unlearning objective via gradient-based methods, as seen in ESD, which adjusts the model so that the score function conditioned on a concept matches the unconditional score function. Closed-form edits derive an explicit update for unlearning, as in UCE, which modifies key and value weights in cross-attention layers to replace concept-specific representations with generic or blank ones. MACE combines both approaches: it uses a closed-form edit to adjust word embeddings in concept-containing prompts and LoRA fine-tuning to suppress concept-related attention in generated images. We categorize ESD and SDD as fine-tuning methods, UCE as closed-form, and MACE as a hybrid approach.

**Finetuning vs. Closed-Form** In Table 1, we see a gap in the severity of concept resurgence between the fine-tuning algorithms and those using closed-form edits. Specifically, UCE is quite robust, exhibiting very small resurgence across tasks. We conjecture that UCE is the strongest type of closed-form edit, as it modifies the cross attention weights to directly map the target concept to a higher-level (more abstract) concept. For example, if the target concept is a particular celebrity, it may be mapped to the more abstract concept like "a Person" or "a Celebrity". In contrast, MACE modifies the cross-attention weights to map the embeddings of all the surrounding words in the given prompts to be similar to embeddings of the surrounding words after replacing the target concept with a more abstract one. This difference means that MACE does not directly optimize the parameter update to move the target concept embedding towards the abstract concept embedding. Furthermore, because MACE incorporates unlearning the target concept information via fine-tuning, this might leave it more vulnerable to concept resurgence than UCE, which is based on a direct closed-form edit.

**Parameter Choice** The second algorithmic factor we examine is which subsets of parameters are updated in the unlearning phase, and which (potentially overlapping) subsets of parameters are further fine-tuned. We start by showing how these choices potentially explain why UCE is more robust to concept resurgence than the other three algorithms. As discussed above, UCE only modifies the cross-attention weights with a closed form edit. As discussed in (16), this approach is very effective for concepts that are localized to the words themselves (e.g. the name of a celebrity; contrast this to unsafe content, which is a more abstract concept). Applying LoRA fine-tuning after UCE unlearning, we find no evidence of concept resurgence. We then instead fine-tune the full set of parameters, which yields a small degree of resurgence. Finally, motivated by this result, we choose to fully fine-tune the cross-attention layers only. We see that the resurgence is comparable between the two (Table 3), suggesting that the nature of UCE's closed-form edit being localized to the cross-attention layers may make it very robust.

The second difference between the seven algorithms is the subset of model parameters that are updated in the unlearning step. Section 3 focuses primarily on modifying the either the cross-attention layers (with the exception of MACE, which also updates the rest of the model parameters via LoRA fine-tuning) or the automatically selected parameter subset (i.e. SalUn and SHS). Here, we focus on ESD in the single celebrity erasure task and the copyright erasure task, which both exhibit very high degrees of concept resurgence. In each of these tasks, we vary the subset of parameters that are updated in the unlearning step: either all of the parameters, all of the parameters except those in the cross-attention layers, and only those in the cross-attention layers. We find that the cross-attention parameters do indeed play the most important role in unlearning for these tasks and that unlearning on all the parameters only provided marginal gains in preventing resurgence (Figure 16).

**Finetuning Hyperparameters** Finally, we investigate how hyperparameter choices such as dataset size and number of fine-tuning steps impact the severity of resurgence. In Appendix J, we show that even with much smaller amounts of data or smaller amounts of fine-tuning steps that resurgence still occurs. For example, for MACE on our erase 10 celebrities task, only 20 fine-tuning steps are necessary for resurgence to occur with 250 samples.

# 5 Why Does Concept Resurgence Occur?

We argue that concept resurgence is a structural vulnerability inherent in the geometric relationship between the erased concept and the subsequent fine-tuning task. To isolate the conditions under which forgotten concepts can resurface during fine-tuning, we analyze a *linear score-based diffusion model*. This simplified, convex setting allows the geometry of unlearning and fine-tuning to be made explicit. In particular, we show that *any non-zero overlap* between the subspace associated with an erased concept and the subspace spanned by fine-tuning gradients is sufficient to induce concept resurgence. Let $W$ denote the parameters of the linear score model (vectorized for notational convenience). Unlearning is modeled as an explicit removal of parameter components associated with a target concept. Formally, let $\mathcal{C} \subset \mathbb{R}^d$ denote the *erased concept subspace*, defined as the span of parameter directions that encode the concept prior to unlearning. After unlearning, the parameters satisfy $P_{\mathcal{C}}W \approx 0$, where $P_{\mathcal{C}}$ denotes the orthogonal projection onto $\mathcal{C}$. Let $\mathcal{D}_{\mathrm{FT}}$ denote the fine-tuning dataset, and let $\mathcal{S} \subset \mathbb{R}^d$ denote the subspace spanned by the per-example gradients of the fine-tuning loss evaluated at the post-unlearning parameters $W$, i.e., $\mathcal{S} := \mathrm{span}\left\{\nabla_W \ell(W; x) : x \in \mathcal{D}_{\mathrm{FT}}\right\}$. Crucially, $\mathcal{S}$ is a *gradient-induced subspace*: it is determined jointly by the data, the loss, and the current model state, and does not, in general, coincide with the full parameter space even under nominally full fine-tuning.

Our central structural assumption is that these two subspaces are not orthogonal, $P_{\mathcal{C}}(\mathcal{S}) \neq \{0\}$. This condition is mild and generically satisfied in high-dimensional models. Even when the fine-tuning task is conceptually unrelated to the erased concept, the induced gradient directions will generically exhibit incidental alignment with previously erased directions. In the linear model with a quadratic loss, this overlap condition is sufficient to guarantee resurgence: fine-tuning generically reintroduces non-zero components of $W$ along $\mathcal{C}$. For nonlinear models, overlap should instead be interpreted as a *structural precondition* that makes resurgence possible, but not inevitable. In practice, the magnitude and observability of resurgence are mediated by nonlinearity, curvature, and gradient redistribution effects, as reflected in the differing behaviors observed under full versus constrained fine-tuning regimes (e.g., UCE-All versus restricted updates). While overlap explains why resurgence can occur, it does not explain why it can become substantial. Our goal is therefore to characterize how this weak geometric interaction can be amplified into meaningful resurgence in our simplified linear setting. In particular, our analysis identifies two bounds that govern how erased components grow under fine-tuning in this model

- *Gradient resurgence bound.* This bound identifies when fine-tuning gradients reappear in the forgotten subspace $\mathcal{C}$, despite prior unlearning. It shows that nonzero gradient mass arises in $\mathcal{C}$ whenever there is residual alignment between the fine-tuning subspace $\mathcal{S}$ and $\mathcal{C}$. Formally:

$$\|P_{\mathcal{C}}\left(\nabla_W \mathcal{L}_t\right)\|_F \geq 2\sqrt{1-\alpha_t} \cdot \sqrt{\gamma(\mathcal{S}, \mathcal{C})},$$

  where $\gamma(\mathcal{S}, \mathcal{C}) \triangleq \lambda_{\min}(P_{\mathcal{S}}P_{\mathcal{C}}P_{\mathcal{S}})$ measures the worst-case leakage from $\mathcal{S}$ into $\mathcal{C}$. This overlap ensures that even when concepts in $\mathcal{C}$ have been suppressed, fine-tuning gradients computed from noise-perturbed data can reintroduce them if they are not fully orthogonal to the directions encoded in the new task. Notably, this bound is most active at early diffusion timesteps, where $1 - \alpha_t$ is large and thus amplifies the residual error when there is *any* amount of overlap.

- *Curvature-limited sensitivity.* This bound captures the model's geometric sensitivity to reactivation. Even if gradient mass in $\mathcal{C}$ is small, the induced update can be large if the curvature in those directions is low. Formally, for any update $\Delta W$ supported in $\mathcal{C}$, we have:

$$\|P_{\mathcal{C}}\Delta W\|_F \geq \frac{2\sqrt{1-\alpha_t} \cdot \sqrt{\gamma(\mathcal{S}, \mathcal{C})}}{\alpha_t \lambda_{\max}^{\mathcal{C}} + (1-\alpha_t)},$$

  where $\lambda_{\max}^{\mathcal{C}} := \lambda_{\max}(P_{\mathcal{C}}\Sigma P_{\mathcal{C}})$ is the maximum variance in the forgotten subspace. This bound reveals a key amplification mechanism: low-variance directions are highly sensitive to reactivation, since small gradients can produce large updates when curvature is shallow.

**Proposition 5.1** (Linear diffusion model resurgence). *Assume a linear diffusion model with residual of the form*

$$\epsilon_W(x_t, t) := W x_t - \epsilon$$

*for some matrix $W \in \mathbb{R}^{d \times d}$, where $\epsilon \sim \mathcal{N}(0, I)$ is independent Gaussian noise. Let $\mathcal{C} \subset \mathbb{R}^d$ be a subspace, and let $\mathcal{D}_{\mathrm{FT}}$ be a fine-tuning dataset whose induced gradient directions span a subspace $\mathcal{S}$. Let $P_{\mathcal{C}} = U_{\mathcal{C}} U_{\mathcal{C}}^{\top}$ and $P_{\mathcal{S}} = U_{\mathcal{S}} U_{\mathcal{S}}^{\top}$ denote the orthogonal projection matrices onto $\mathcal{C}$ and $\mathcal{S}$, respectively. Define the leakage*

$$\gamma(\mathcal{S}, \mathcal{C}) \triangleq \lambda_{\min}(P_{\mathcal{S}} P_{\mathcal{C}} P_{\mathcal{S}}).$$

*Let $x_0 \sim \mathcal{D}_{\mathrm{FT}}$ have covariance $\Sigma$, and define the forward-corrupted input as*

$$x_t := \sqrt{\alpha_t} x_0 + \sqrt{1 - \alpha_t} \epsilon \quad \text{so that} \quad \Sigma_t := \mathbb{E}[x_t x_t^{\top}] = \alpha_t \Sigma + (1 - \alpha_t) I.$$

*Let $\lambda_{\max}^{\mathcal{C}} \triangleq \lambda_{\max}(P_{\mathcal{C}} \Sigma P_{\mathcal{C}})$ and suppose $P_{\mathcal{C}} \mathcal{S} \neq 0$. Then if the prior unlearning was successful, i.e. $P_{\mathcal{C}} W = 0$, we obtain bounds characterizing resurgence:*

1. **Gradient resurgence:** *The fine-tuning gradient projected into $\mathcal{C}$ satisfies:*

$$\|P_{\mathcal{C}} (\nabla_W \mathcal{L}_t)\|_F \geq 2\sqrt{1 - \alpha_t} \cdot \sqrt{\gamma(\mathcal{S}, \mathcal{C})}.$$

2. **Curvature-limited sensitivity:** *The update $\Delta W \in \mathbb{R}^{d \times d}$ in the weight matrix supported in the forgotten subspace $\mathcal{C}$ satisfies:*

$$\|P_{\mathcal{C}} \Delta W\|_F \geq \frac{2\sqrt{1 - \alpha_t} \cdot \sqrt{\gamma(\mathcal{S}, \mathcal{C})}}{\alpha_t \lambda_{\max}^{\mathcal{C}} + (1 - \alpha_t)}.$$

We provide a proof of the gradient resurgence bound in Appendix G and a proof of the curvature-limited sensitivity bound in Appendix H. The basic idea behind the gradient bound is to observe that the norm of the fine-tuning gradient projected onto the erased subspace $\mathcal{C}$ is lower bounded by the Frobenius norm of the matrix $A := \mathbb{E}[\epsilon_W(x_t, t) x_t^{\top}]$ restricted to $\mathcal{C}$, multiplied by the overlap term $\gamma(\mathcal{S}, \mathcal{C})$. When exact unlearning has occurred, i.e., when $P_{\mathcal{C}} W = 0$, this expression simplifies to a term proportional to $1 - \alpha_t$, reflecting the contribution of noise-perturbed inputs in the diffusion process. The corresponding bound on the induced parameter update exploits the fact that, in this linear setting, the loss is quadratic in $W$, allowing the update magnitude to be controlled by the inverse curvature along $\mathcal{C}$.

Taken together, these bounds clarify the structural and dynamical factors that govern concept resurgence after unlearning. First, the gradient resurgence bound characterizes when fine-tuning gradients reappear in the erased subspace $\mathcal{C}$. The bound depends explicitly on the overlap between $\mathcal{C}$ and the fine-tuning gradient subspace $\mathcal{S}$, as quantified by the leakage term $\gamma(\mathcal{S}, \mathcal{C})$, and scales with $\sqrt{1 - \alpha_t}$, capturing the increased influence of noise-perturbed inputs at early diffusion timesteps. In the idealized setting where $P_{\mathcal{C}} W = 0$, this term isolates resurgence driven purely by residual alignment between the new task gradients and previously erased directions. When unlearning is imperfect and signal remains in $W$, an additional contribution proportional to $P_{\mathcal{C}} W \Sigma$ appears, which can dominate at later timesteps and further amplify resurgence. Second, the curvature-limited sensitivity bound governs how strongly the model responds to gradient mass in $\mathcal{C}$. Even when the projected gradient is small, the induced parameter update can be large if the loss curvature in those directions is shallow. This amplification is most pronounced when $\mathcal{C}$ aligns with low-variance directions in the data, corresponding to small $\lambda_{\max}^{\mathcal{C}}$, and when $\alpha_t$ is large but not too close to one, so that noise has diminished while curvature remains anisotropic. In contrast, at early timesteps where $\alpha_t \ll 1$, the effective curvature is nearly isotropic, suppressing updates and making reactivation less likely.

This analysis also provides intuition for the empirical behavior observed under different fine-tuning strategies. Restricting fine-tuning to a small subset of parameters concentrates gradient energy into fewer directions, increasing effective overlap with erased subspaces and exposing low-curvature directions that amplify reactivation. In contrast, broader or full fine-tuning allows gradients to redistribute across the parameter space and enables the model to adjust curvature more globally, reducing both gradient leakage into $\mathcal{C}$ and its subsequent amplification. This perspective explains why, despite the generic presence of overlap between $\mathcal{C}$ and $\mathcal{S}$, full fine-tuning can empirically mitigate resurgence, as observed in Figure 16 and in the ESD parameter search.

## 6 Discussion and Limitations

Our investigation opens several important directions for future work. First, our theoretical analysis is restricted to the linear setting, and it remains an open question whether similar characterizations of concept resurgence extend to nonlinear models. Exploring such extensions could inform new strategies for mitigating resurgence and improving the robustness of unlearning procedures. Second, our empirical evaluation is limited to standard academic benchmarks and synthetic settings. Further research is needed to assess the practical relevance of concept resurgence in real-world deployments, particularly in scenarios involving long-horizon or compositional fine-tuning, where interleaved updates may amplify vulnerabilities.

Concept resurgence also raises important questions about responsibility for downstream harms. Despite a developer's best efforts to sanitize a model using these techniques, a downstream user who fine-tunes a published model might be surprised to discover that guardrails put in place by the developer no longer exist. This creates a dilemma: is the developer obligated to permanently and irrevocably erase problematic concepts, or does responsibility shift to the downstream if they (inadvertently) reintroduce them? Despite these challenges, concept unlearning remains a valuable tool for model developers. By identifying its vulnerabilities, our work aims to drive the development of erasure techniques that remain robust throughout a model's life-cycle or develop tools that can help developers anticipate when concept resurgence is likely to happen.

## 7 Acknowledgements

VMS was supported by the Bridgewater AIA Labs Research Fellowship.

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

# A    Additional Qualitative Examples

In this section, we include qualitative results for the copyright and object erasure tasks in Figure 4 and Figure 5, respectively. These results are analogous to those presented in Figure 2 for the celebrity erasure task. We choose to exclude qualitative examples of resurgence for the unsafe content task, as these may be upsetting. For a quantitative evaluation of this task across all seven unlearning algorithms, we refer readers to Table 1.

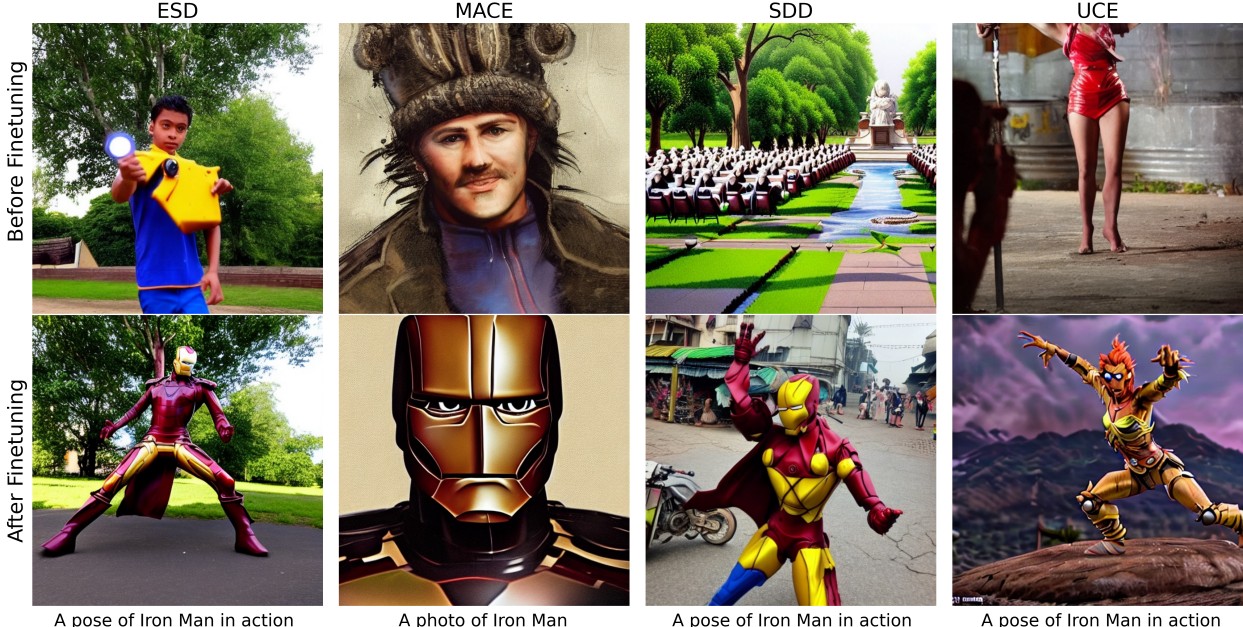

Figure 4: Selected images generated by SD v1.4 after initially applying each unlearning algorithm (top row) and after subsequent fine-tuning (bottom row) in the copyright unlearning task. In each case, the model initially unlearns the target concept; in this case, how to generate images of Iron Man. However, fine-tuning on unrelated images can inadvertently reintroduce the erased concept.

# B    Unlearning Tasks

For the copyright task, we choose the concept "Iron Man" to erase. We generate five prompts that we provide the model with five different random seeds to evaluate its knowledge of Iron Man. These prompts were:

1. "A photo of [name]"

2. "A portrait of [name]"

3. "A pose of [name] in action"

4. "An illustration of [name]"

5. "An iconic scene of [name]"

Additionally, we create retain set of copyright characters that include: Albus Dumbledore, Anna, Aquaman, Aragorn, Arwen, Barbie, Bart Simpson, Batman, Black Panther, Black Widow, Boromir, Bugs Bunny, Buzz Lightyear, C-3PO, Captain America, Catwoman, Chewbacca, Daffy Duck, Darth Vader, Doctor Strange, Donald Duck, Darth Vader, Doctor Strange, Donald Duck, Donkey, Dr. Watson, Draco Malfoy, Dracula, Ebenezer Scrooge, Elsa Mars, Elsa, Fiona, Flash, Frankenstein's Monster, Fred Flinstone, Frodo Baggins,

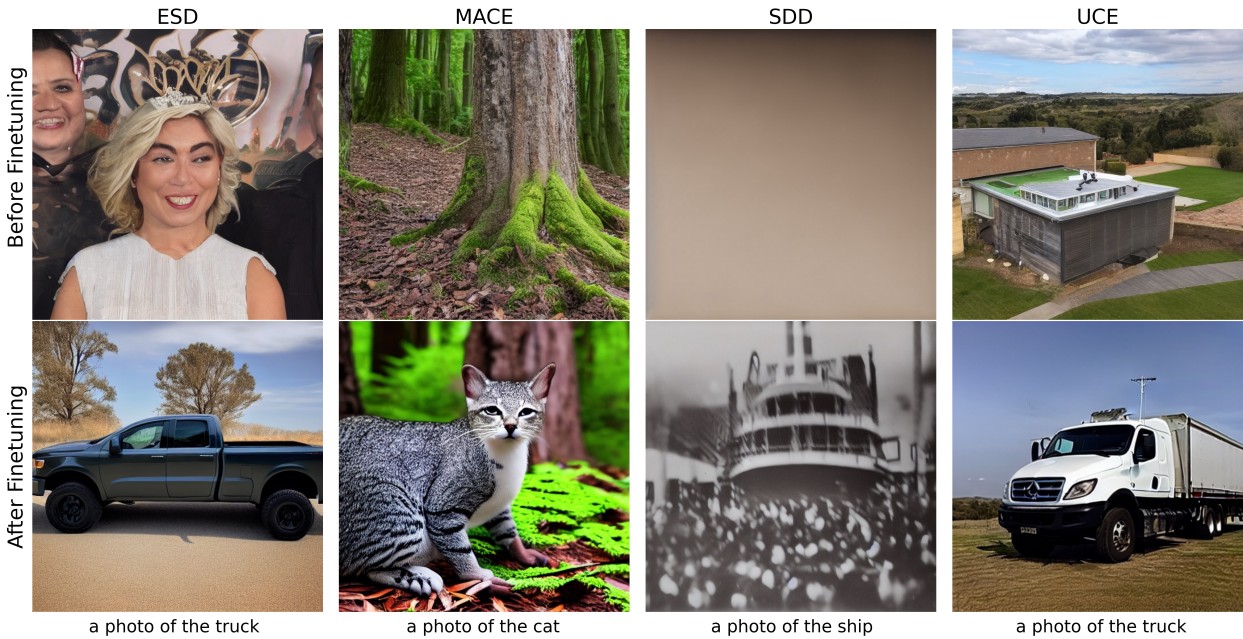

Figure 5: Selected images generated by SD v1.4 after initially applying each unlearning algorithm (top row) and after subsequent fine-tuning (bottom row) in the object unlearning task. In each case, the model initially unlearns the target concept; e.g., how to generate images of a truck. However, fine-tuning on unrelated images can inadvertently reintroduce the erased concepts.

Galadriel, Gandalf, Gollum, Goofy, Green Lantern, Hagrid, Han Solo, Harley Quinn, Harry Potter, Hermione Granger, Homer Simpson, Huckleberry Finn, Hulk, Jack Sparrow, Joker, Juliet, Katniss Everdeen, Kirby, Kylo Ren, Lara Croft, Legolas, Lex Luthor, Link, Loki, Luigi, Luke Skywalker, Luna Lovegood, Mario, Master Chief, Mickey Mouse, Minnie Mouse, Moana, Neo, Neville Longbottom, Obi-Wan Kenobi, Oliver Twist, Patrick Star, Peter Griffin, Pikachu, Princess Leia, Princess Peach, R2D2, Romeo, Ron Weasley, Samwise Gamgee, Sauron, Scarlet Witch, Scooby-Doo, Severus Snape, Shaggy, Sherlock Holmes, Shrek, Simba, Snoopy, Sonic the Hedgehog, Spider-Man, Spongebob Squarepants, Superman, Thor, Tom Sawyer, Tony Montana, Voldemort, Willy Wonka, Wonder Woman, Woody, and Yoda.

For the unsafe content task, we select a subset of concepts from the Inappropriate Images Prompts (I2P) (41) dataset. We are focused on erasing the concepts hate, self-harm, violence, and harassment. We select prompts labeled as such in the dataset and that have a score of at least 70% or more on the Q16 percentage. This percentage represents how many times out of 10 samples the Q16 classifier classified the image as inappropriate.

## C   Fine-tuning Dataset Curation and Training Details

In this section we provide additional details related to the dataset curation process for the different tasks. The "random" dataset for celebrities, includes 25 images of 10 distinct celebrities, chosen arbitrarily from those used in (32) while ensuring that they do not overlap with any of the erased celebrities in any of the subtasks. These celebrities are Amy Winehouse, Elizabeth Taylor, George Takei, Henry Cavill, Jeff Bridges, Jensen Ackles, Jimmy Carter, Kaley Cuoco, Kate Upton and Kristen Stewart. For each celebrity, we generated five images for each of five prompts (25 total). These prompts were:

1. "A portrait of [name]"

2. "An image capturing [name] at a public event"

3. "A sketch of [name]"

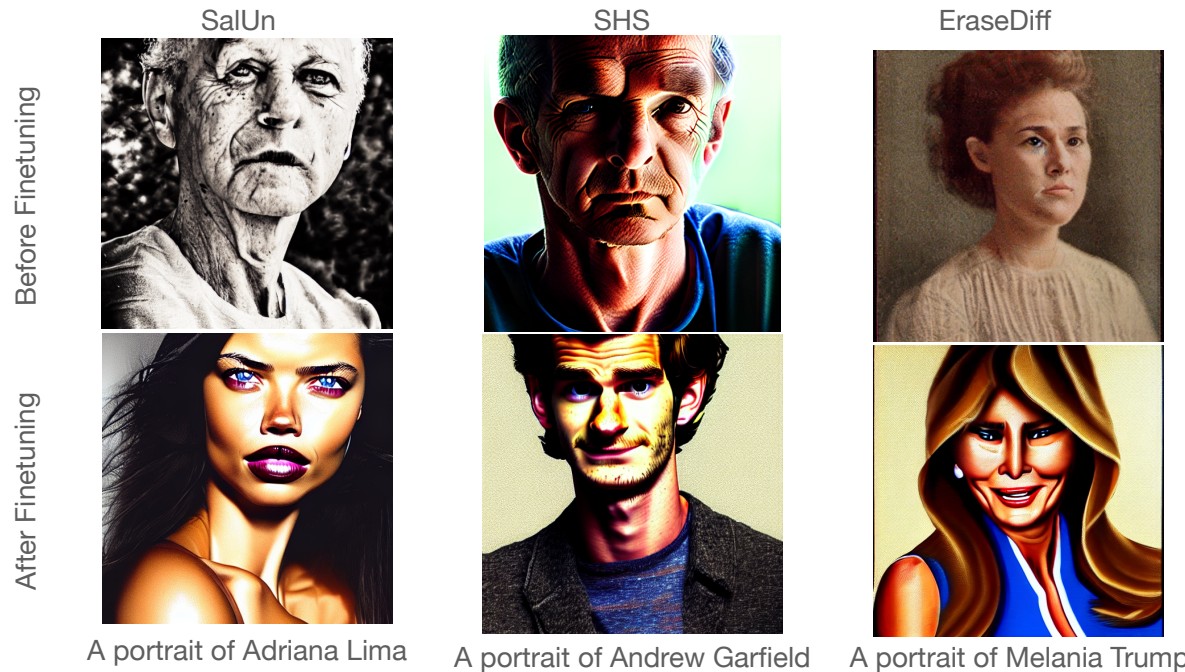

Figure 6: Selected images generated by SD v1.4 after initially applying each new unlearning algorithm (top row) and after subsequent fine-tuning (bottom row) in the celebrity unlearning task. In each case, the model initially unlearns the target concept; e.g., how to generate images of Andrew Garfield. However, fine-tuning on unrelated images still inadvertently reintroduces the erased concepts on these new baselines.

    4. "An oil painting of [name]"

    5. "[name] in an official photo"

The "random" dataset for objects, includes 5 images of 8 distinct objects, chosen arbitrarily from the classes of CIFAR-100 (28) while ensuring that they do not overlap with any of the erased objects. These objects are trout, ray, bee, rose, lobster, girl, oak tree, aquarium fish, Kate Upton and Kristen Stewart. For each object, we generated five images for each prompt. The prompt used was "a photo of the [object]."

The "random" dataset for copyright includes 5 images of different concepts chosen from the retain set described in Appendix B with the prompts:

    1. "A photo of [name]"

    2. "A portrait of [name]"

    3. "A pose of [name] in action"

    4. "An illustration of [name]"

    5. "An iconic scene of [name]"

The characters chosen for fine-tuning are Shaggy, Simba, Daffy Duck, Spongebob Squarepants, Luigi, Arwen, Galadriel, Gandalf, and Hagrid.

Finally, the "random" dataset for unsafe concepts takes the prompts from the i2p dataset that are labeled as 0% on the Q16 percentage score meaning out of 10 samples they were never classified as inappropriate from Q16.

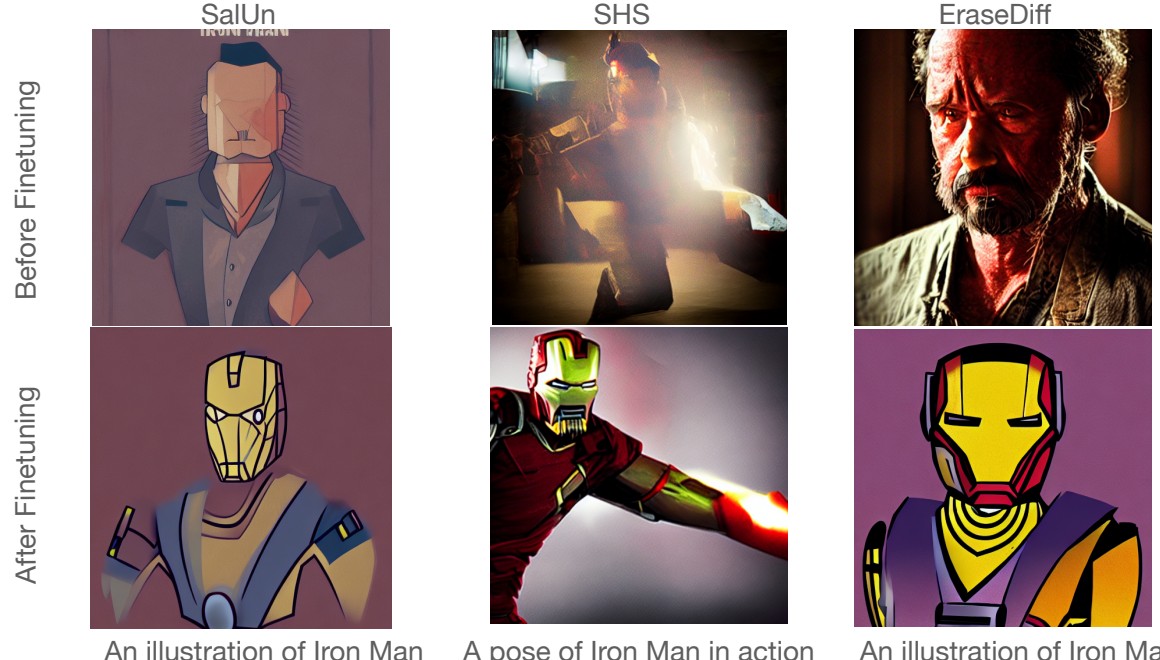

Figure 7: Selected images generated by SD v1.4 after initially applying each new unlearning algorithm (top row) and after subsequent fine-tuning (bottom row) in the copyright unlearning task. In each case, the model initially unlearns the target concept; e.g., how to generate images of Iron Man. However, fine-tuning on unrelated images still inadvertently reintroduces the erased concepts on these new baselines.

**Training Details**   We perform finetuning for each task on each of these datasets described above using LoRA (unless otherwise specified – e.g. for UCE we apply full parameter fine-tuning) for 1000 steps. The size of the fine-tuning datasets varies based on the task, details are above. We experiment with these parameters for MACE in Appendix J, showing that resurgence occurs even if we reduce the number of finetuning steps.

## D   Stable Diffusion 2.1 Results

In this section we present results which are analogous to those in Table 1 for Stable Diffusion v2.1.

## E   Additional Scaling Analyses

In this section we present additional results illustrating the degree of concept resurgence for SDD, MACE, UCE, SalUn, SHS, and EraseDiff as the number of erased concepts grows in the celebrity and object erasure tasks. These results are presented in Figure 10, Figure 11, Figure 12, Figure 15, Figure 14, Figure 13 respectively, and are analogous to the results presented in Figure 9 for the ESD algorithm.

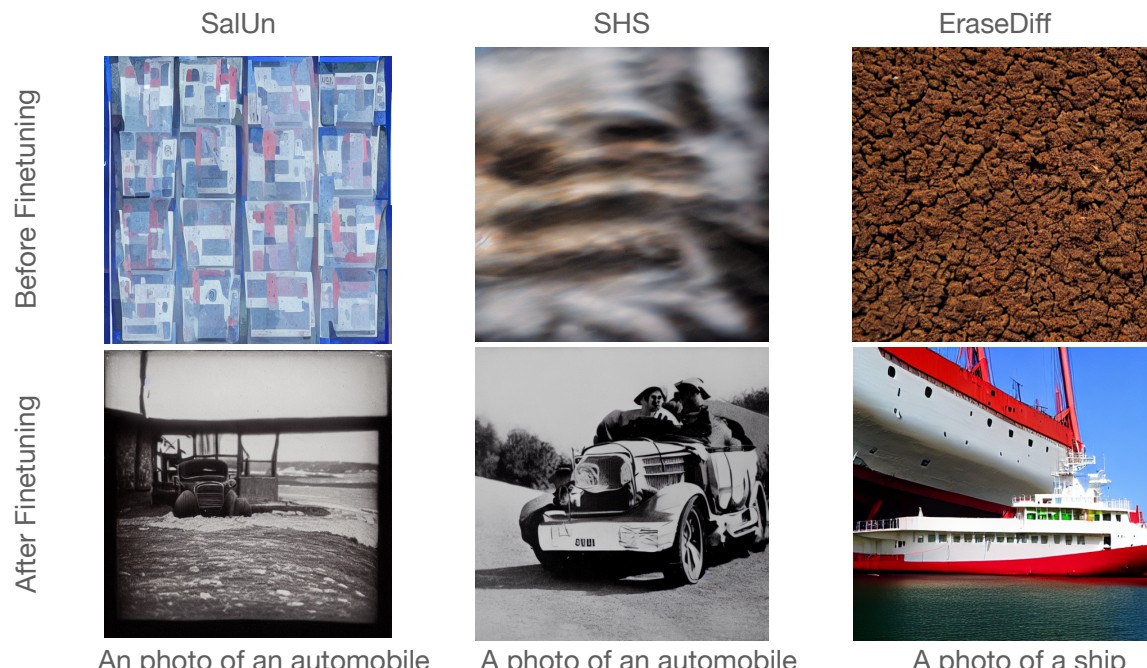

Figure 8: Selected images generated by SD v1.4 after initially applying each new unlearning algorithm (top row) and after subsequent fine-tuning (bottom row) in the object unlearning task. In each case, the model initially unlearns the target concept; e.g., how to generate images of an automobile. However, fine-tuning on unrelated images still inadvertently reintroduces the erased concepts on these new baselines.

Table 2: Unlearning performance before and after fine-tuning for Stable Diffusion v2.1. Each metric is task-specific, and evaluates the ability to generate the unwanted concept (lower is better; see Section 3 for details). Results for SDD on unsafe content are excluded as first-stage unlearning compromises the model's ability to generate *any* images, including retained concepts.

| Task | Algorithm | Before FT | After FT |
|---|---|---|---|
| **celebrity** | **ESD** | $0.291 \pm 0.095$ | $0.929 \pm 0.011$ |
| | **SDD** | $0.804 \pm 0.087$ | $0.934 \pm 0.023$ |
| | **UCE** | $0.002 \pm 0.000$ | $0.004 \pm 0.001$ |
| **copyright** | **ESD** | $0.000 \pm 0.000$ | $0.000 \pm 0.033$ |
| | **SDD** | $0.000 \pm 0.000$ | $0.167 \pm 0.100$ |
| | **UCE** | $0.000 \pm 0.000$ | $0.000 \pm 0.000$ |
| **unsafe** | **ESD** | $0.155 \pm 0.023$ | $0.780 \pm 0.013$ |
| | **SDD** | N/A | N/A |
| | **UCE** | $0.652 \pm 0.000$ | $0.715 \pm 0.021$ |

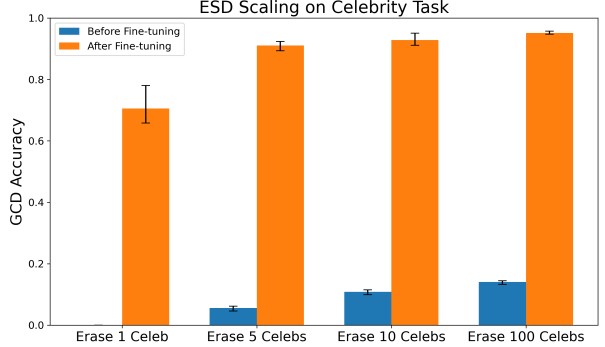 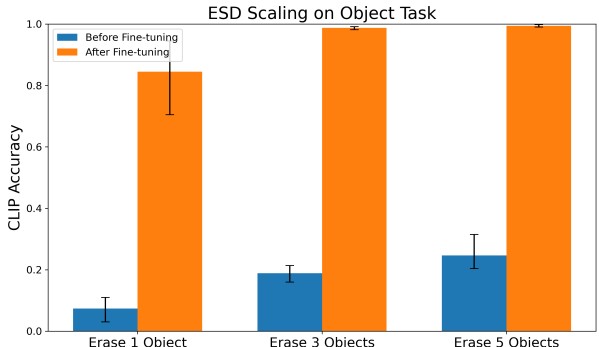

(a) Scaling the ESD algorithm to erase multiple celebrities    (b) Scaling the ESD algorithm to erase multiple objects

Figure 9: Quantifying the severity of concept resurgence as the number of erased concepts increases for the ESD algorithm. As the unlearning task becomes more challenging, the degree of concept resurgence increases sharply.

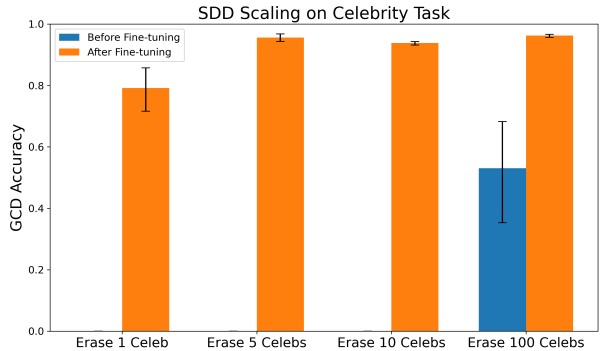 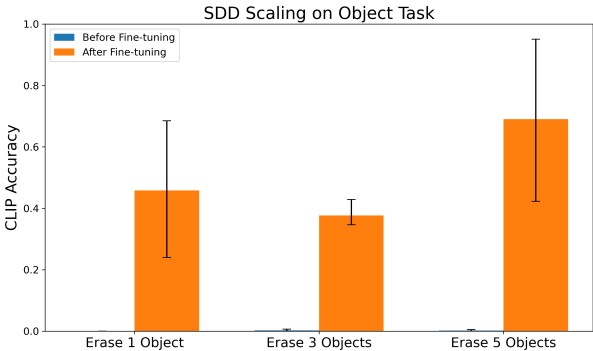

(a) Scaling the SDD algorithm to erase multiple celebrities    (b) Scaling the SDD algorithm to erase multiple objects

Figure 10: Quantifying the severity of concept resurgence as the number of erased concepts increases for the SDD algorithm.

## F  Additonal Algorithm Choice Analyses

In this section we present additional results illustrating the algorithmic choices for ESD and UCE that impact resurgence.

| Method | Before FT | After X-Attn FT | After Full FT |
|---|---|---|---|
| Erase 5 | 0.000 (0.000, 0.000) | 0.004 (0.004, 0.004) | 0.001 (0.000, 0.004) |
| Erase 10 | 0.004 (0.004, 0.004) | 0.004 (0.000, 0.008) | 0.000 (0.000, 0.000) |
| Erase 100 | 0.001 (0.001, 0.001) | 0.001 (0.001, 0.001) | 0.003 (0.002, 0.004) |

Table 3: Comparison of fine-tuning different subsets of parameters after UCE unlearning across different erase celebrity subtasks. Full fine-tuning of just cross-attention layers provides comparable resurgence to full fine-tuning of all parameters.

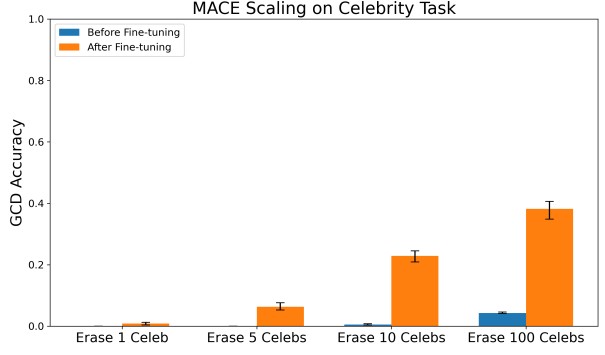

(a) Scaling the MACE algorithm to erase multiple celebrities

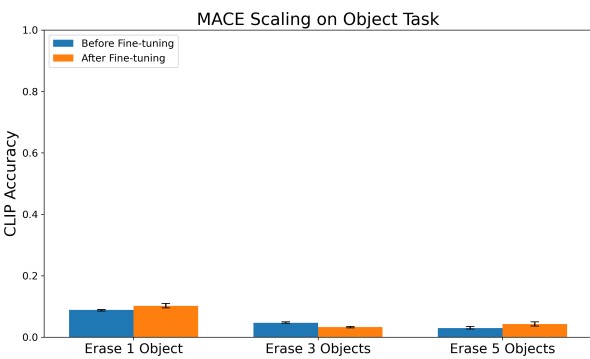

(b) Scaling the MACE algorithm to erase multiple objects

Figure 11: Quantifying the severity of concept resurgence as the number of erased concepts increases for the MACE algorithm.

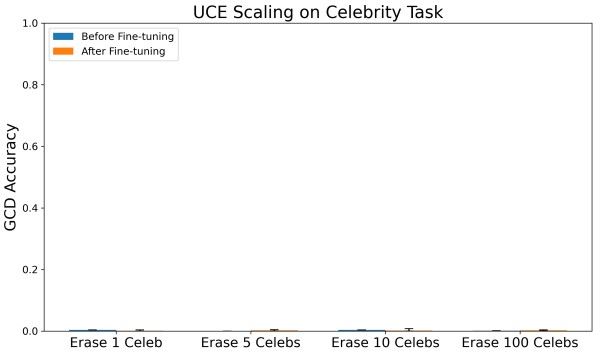

(a) Scaling the UCE algorithm to erase multiple celebrities

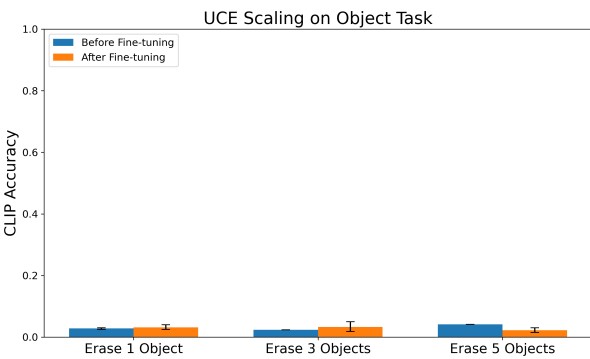

(b) Scaling the UCE algorithm to erase multiple objects

Figure 12: Quantifying the severity of concept resurgence as the number of erased concepts increases for the UCE algorithm. As the left panel demonstrates, UCE is highly robust to resurgence on all four of the celebrity erasure tasks.

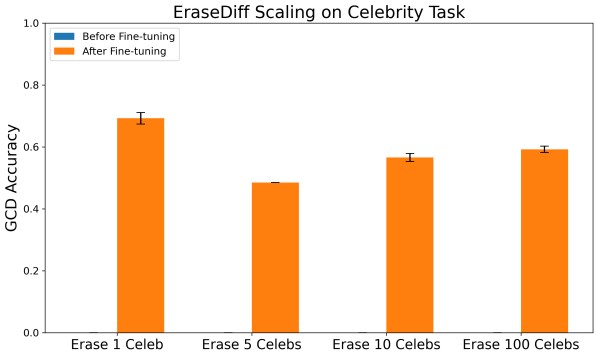

(a) Scaling the EraseDiff algorithm to erase multiple celebrities

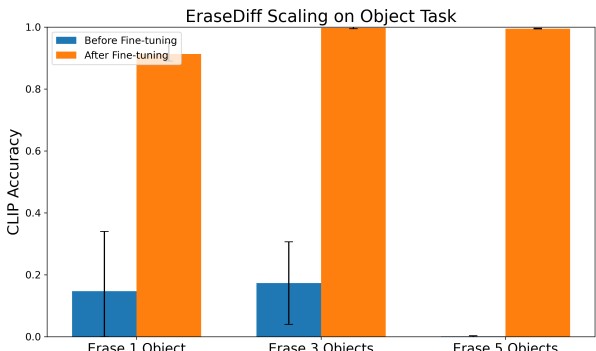

(b) Scaling the EraseDiff algorithm to erase multiple objects

Figure 13: Quantifying the severity of concept resurgence as the number of erased concepts increases for the EraseDiff algorithm.

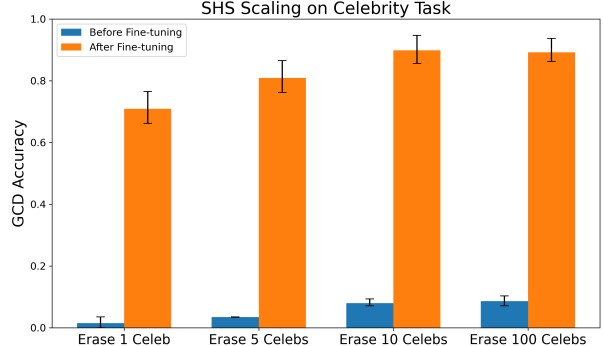
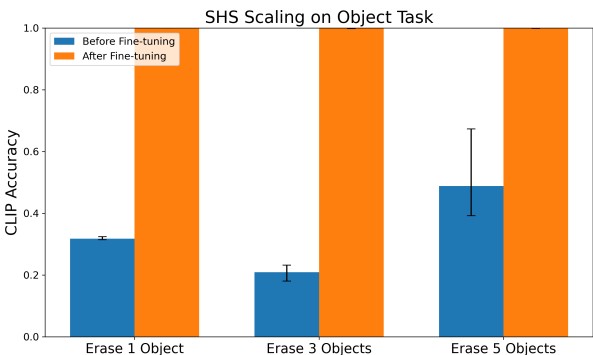

(a) Scaling the SHS algorithm to erase multiple celebrities

(b) Scaling the SHS algorithm to erase multiple objects

Figure 14: Quantifying the severity of concept resurgence as the number of erased concepts increases for the SHS algorithm.

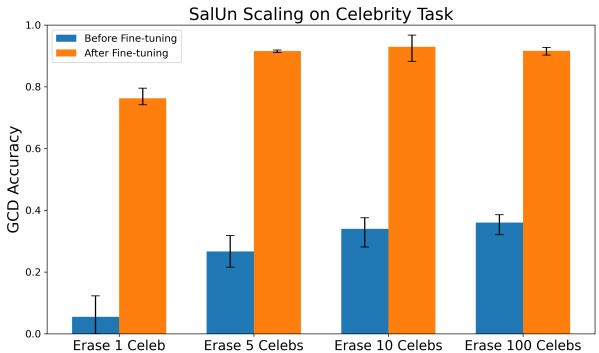
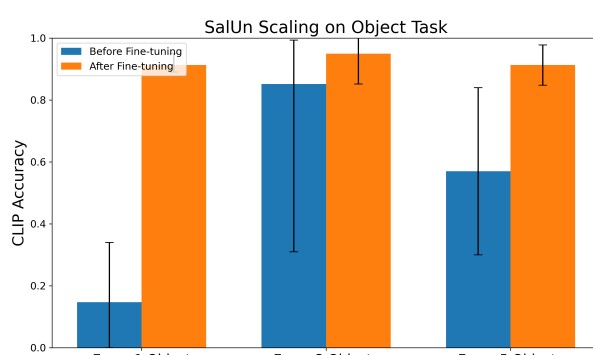

(a) Scaling the SalUn algorithm to erase multiple celebrities

(b) Scaling the SalUn algorithm to erase multiple objects

Figure 15: Quantifying the severity of concept resurgence as the number of erased concepts increases for the SalUn algorithm.

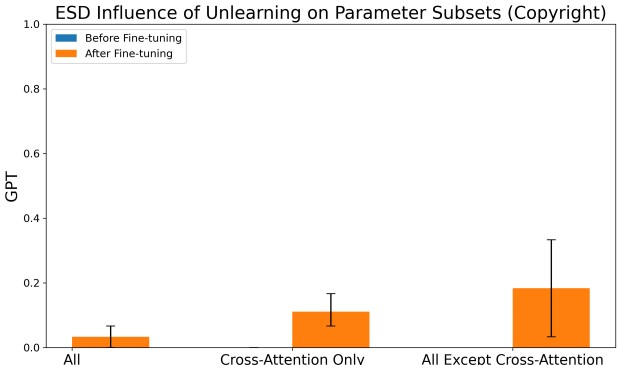

Figure 16: Quantifying the impact of performing unlearning on different subsets of the parameters for the ESD algorithm. Unlearning applied to the cross attention layers helps reduce resurgence and unlearning all on all the parameters helps further.

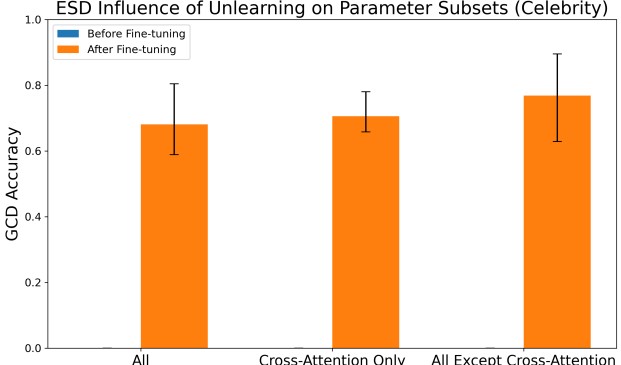

Figure 17: Quantifying the impact of performing unlearning on different subsets of the parameters for the ESD algorithm. Unlearning applied to the cross attention layers helps reduce resurgence and unlearning all on all the parameters helps further.

# G    Proof of Gradient Resurgence Bound

*Proof.* We start with the first bound. Let $A := \mathbb{E}[\epsilon_W(x_t, t)x_t^\top]$. Then:

$$\nabla_W \mathcal{L}_t = 2A, \quad \Rightarrow \quad P_{\mathcal{C}} \nabla_W \mathcal{L}_t = 2P_{\mathcal{C}}A.$$

Applying a standard projection inequality formalized in Lemma G.1 in Appendix G.1, we have

$$\|P_{\mathcal{C}}A\|_F^2 \geq \gamma(\mathcal{S}, \mathcal{C}) \cdot \|P_{\mathcal{S}}A\|_F^2, \quad \Rightarrow \quad \|P_{\mathcal{C}}\nabla_W \mathcal{L}_t\|_F \geq 2 \cdot \sqrt{\gamma(\mathcal{S}, \mathcal{C})} \cdot \|P_{\mathcal{S}}A\|_F.$$

Now we lower bound $\|P_{\mathcal{S}}A\|_F$ via:

$$\|P_{\mathcal{S}}A\|_F \geq \frac{\|A\|_F}{\|P_{\mathcal{S}}^\dagger\|_{\mathrm{op}}} \geq \|A\|_F,$$

since $P_{\mathcal{S}}$ is an orthogonal projection with operator norm 1 so $P_{\mathcal{S}}^\dagger P_{\mathcal{S}}A = A \rightarrow \|A\|_F = \|P_{\mathcal{S}}^\dagger P_{\mathcal{S}}A\|_F \leq \|P_{\mathcal{S}}^\dagger\|_{\mathrm{op}}\|P_{\mathcal{S}}A\|_F$. To lower bound $\|A\|_F$, we consider its directional action. For any unit vector $v \in \mathcal{C}, \|v\|_2 = 1$:

$$\|A\|_F = |\langle A, vv^\top \rangle| = |\mathbb{E}[\langle \epsilon_W(x_t, t), v \rangle \cdot \langle x_t, v \rangle]| = \sqrt{1 - \alpha_t}.$$

This bounds follows from Lemma G.2 which assumes $P_{\mathcal{C}}W = 0$. Details can be found in Appendix G.1. Putting the pieces together, we obtain the first bound. $\qquad\square$

## G.1 Proof of supporting Lemma G.1

**Lemma G.1.** *Let $\mathcal{S}, \mathcal{C} \subseteq \mathbb{R}^d$, be closed linear subspaces with projection matrices $P_\mathcal{S}, P_\mathcal{C} \in \mathbb{R}^{d \times d}$. Then for $X \in \mathbb{R}^{d \times d}$*

$$\|P_\mathcal{C} X\|_F^2 \geq \lambda_{\min}(P_\mathcal{S} P_\mathcal{C} P_\mathcal{S})\|P_\mathcal{S} X\|_F^2$$

*Proof.* We decompose $X$ into its projection into $\mathcal{S}$ and its orthogonal complement $\mathcal{S}_\perp$:

$$X = P_\mathcal{S} X + P_{\mathcal{S}^\perp} X$$

We then apply the projection $P_\mathcal{C}$:

$$P_\mathcal{C} X = P_\mathcal{C} P_\mathcal{S} X + P_\mathcal{C} P_{\mathcal{S}^\perp} X$$

Since $S$ and $S^\perp$ are orthogonal subspaces, the Frobenius norm satisfies

$$\|P_\mathcal{C} X\|_F = \|P_\mathcal{C} P_\mathcal{S} X\|_F + \|P_\mathcal{C} P_{\mathcal{S}^\perp} X\|_F \geq \|P_\mathcal{C} P_\mathcal{S} X\|_F$$

Let $A := P_\mathcal{S} P_\mathcal{C} P_\mathcal{S}$ which is symmetric and PSD and $B := P_\mathcal{S} X$. Then

$$\|P_\mathcal{C} B\|_F^2 = \text{tr}(B^\top P_\mathcal{C} B) = \text{tr}(X^\top P_\mathcal{S} P_\mathcal{C} P_\mathcal{S} X) = \text{tr}(B^\top A B) = \|A^{1/2} B\|_F^2 \geq \lambda_{\min}(A)\|B\|_F^2$$

Giving us the final bound. $\qquad\square$

## G.2 Proof of supporting Lemma G.2

**Lemma G.2** (Directional correlation of residual with input). *Let $x_t = \sqrt{\alpha_t} x_0 + \sqrt{1 - \alpha_t}\epsilon$, where $x_0 \sim \mathcal{D}_{\text{FT}}$ has covariance matrix $\Sigma$, and $\epsilon \sim \mathcal{N}(0, I)$ is independent of $x_0$. Let $\Sigma_t := \alpha_t \Sigma + (1 - \alpha_t)I$ be the covariance of $x_t$, and define the residual error as $\epsilon_W(x_t, t) := W x_t - \epsilon$. Let $A := \mathbb{E}[\epsilon_W(x_t, t) x_t^\top]$. Then for any unit vector $v \in \mathbb{R}^d$, the following identity holds:*

$$|\langle A, vv^\top \rangle| = \left| v^\top W \Sigma_t v - \sqrt{1 - \alpha_t} \right|.$$

*In particular, if $v \in \mathcal{C}$ for a subspace $\mathcal{C}$ such that $P_\mathcal{C} W = 0$, then:*

$$|\langle A, vv^\top \rangle| = \sqrt{1 - \alpha_t}.$$

*Proof.* We expand the matrix inner product:

$$\langle A, vv^\top \rangle = \text{Tr}(A^\top vv^\top) = v^\top A v = \mathbb{E}\left[\langle \epsilon_W(x_t, t), v \rangle \cdot \langle x_t, v \rangle\right].$$

Recall that $\epsilon_W(x_t, t) = W x_t - \epsilon$, so:

$$\mathbb{E}\left[\langle \epsilon_W(x_t, t), v \rangle \cdot \langle x_t, v \rangle\right] = \mathbb{E}\left[(v^\top W x_t)(v^\top x_t) - (v^\top \epsilon)(v^\top x_t)\right].$$

We analyze the two terms separately:

**Term 1:** We simply expand:

$$\mathbb{E}[(v^\top W x_t)(v^\top x_t)] = v^\top W \cdot \mathbb{E}[x_t x_t^\top] \cdot v = v^\top W \Sigma_t v.$$

Note that because $v \in \mathcal{C}$, when $W$ carries no signal in the unlearned space $\mathcal{C}$, i.e. $P_\mathcal{C} W = 0$, then $W^\top v = 0$, hence $v^\top W = 0$, and this term vanishes, simplifying our bound.

**Term 2:** We expand $x_t = \sqrt{\alpha_t} x_0 + \sqrt{1 - \alpha_t}\epsilon$, so:

$$v^\top x_t = \sqrt{\alpha_t} v^\top x_0 + \sqrt{1 - \alpha_t} v^\top \epsilon.$$

Then: $\mathbb{E}[(v^\top \epsilon)(v^\top x_t)] = \sqrt{\alpha_t}\mathbb{E}[(v^\top \epsilon)(v^\top x_0)] + \sqrt{1-\alpha_t}\mathbb{E}[(v^\top \epsilon)^2]$. Since $\epsilon$ is independent of $x_0$ and has zero mean, $\mathbb{E}[(v^\top \epsilon)(v^\top x_0)] = 0$. Also, $v^\top \epsilon \sim \mathcal{N}(0, 1)$, and therefore $\mathbb{E}[(v^\top \epsilon)^2] = 1$. This allows us to conclude

$$\mathbb{E}[(v^\top \epsilon)(v^\top x_t)] = \sqrt{1-\alpha_t}.$$

Putting the two terms together:

$$v^\top A v = v^\top W \Sigma_t v - \sqrt{1-\alpha_t}, \quad \Rightarrow \quad |\langle A, vv^\top \rangle| = \left| v^\top W \Sigma_t v - \sqrt{1-\alpha_t} \right|.$$

$\square$

## H Curvature-Limited Sensitivity Bound

**Lemma H.1** (Lower bound on update magnitude in forgotten subspace). *Let $\mathcal{L}_t(W) = \mathbb{E}_{x_t, \epsilon}\left[ \|W x_t - \epsilon\|^2 \right]$ be the time-t loss for a linear diffusion model, where $x_t \in \mathbb{R}^d$ has covariance $\Sigma_t = \alpha_t \Sigma + (1-\alpha_t)I$ for some positive semidefinite matrix $\Sigma$. Let $\mathcal{C} \subset \mathbb{R}^d$ be a subspace with projection matrix $P_{\mathcal{C}}$, and define*

$$\lambda_{\max}^{\mathcal{C}} := \lambda_{\max}(P_{\mathcal{C}} \Sigma P_{\mathcal{C}}).$$

*Then for any update $\Delta W \in \mathbb{R}^{d \times d}$ satisfying $\Delta W = P_{\mathcal{C}} \Delta W$ (i.e., supported in $\mathcal{C}$), the following holds:*

$$\|P_{\mathcal{C}} \Delta W\|_F \geq \frac{\|P_{\mathcal{C}} \nabla_W \mathcal{L}_t\|_F}{2(\alpha_t \lambda_{\max}^{\mathcal{C}} + (1-\alpha_t))}.$$

*Proof.* We begin by observing that the loss is a quadratic function in $W$, and hence admits a second-order Taylor expansion. For any update $\Delta W \in \mathbb{R}^{d \times d}$, we have:

$$\mathcal{L}_t(W + \Delta W) = \mathcal{L}_t(W) + \langle \nabla_W \mathcal{L}_t, \Delta W \rangle + \frac{1}{2}\langle \Delta W, \nabla_W^2 \mathcal{L}_t[\Delta W] \rangle.$$

We now construct a descent direction in the subspace $\mathcal{C}$. Let $G := P_{\mathcal{C}} \nabla_W \mathcal{L}_t$ denote the gradient projected into $\mathcal{C}$. Consider the update $\Delta W := -\eta G$ for some step size $\eta > 0$. Then $\Delta W \in \mathcal{C}$, and we compute:

$$\mathcal{L}_t(W + \Delta W) - \mathcal{L}_t(W) = -\eta \|G\|_F^2 + \frac{1}{2}\eta^2 \langle G, \nabla_W^2 \mathcal{L}_t[G] \rangle.$$

To ensure descent, we require:

$$-\eta \|G\|_F^2 + \frac{1}{2}\eta^2 \langle G, \nabla_W^2 \mathcal{L}_t[G] \rangle < 0.$$

Define $c := \langle G, \nabla_W^2 \mathcal{L}_t[G] \rangle$. Then the minimum of this quadratic in $\eta$ is achieved at:

$$\eta^* = \frac{\|G\|_F^2}{c}, \quad \text{with} \quad \|\Delta W\|_F = \eta^* \cdot \|G\|_F = \frac{\|G\|_F^3}{c}.$$

We now upper bound the curvature term $c$. Since $\nabla_W^2 \mathcal{L}_t = 2\Sigma_t$, we have:

$$\langle G, \nabla_W^2 \mathcal{L}_t[G] \rangle = 2 \cdot \mathrm{Tr}(G^\top \Sigma_t G) = 2\|G\Sigma_t^{1/2}\|_F^2.$$

Because $G \in \mathcal{C}$, the maximum value of this term is bounded by:

$$\langle G, \nabla_W^2 \mathcal{L}_t[G] \rangle \leq 2 \cdot \lambda_{\max}(P_{\mathcal{C}} \Sigma_t P_{\mathcal{C}}) \cdot \|G\|_F^2.$$

By direct expansion of $\Sigma_t$, we compute:

$$P_{\mathcal{C}} \Sigma_t P_{\mathcal{C}} = \alpha_t P_{\mathcal{C}} \Sigma P_{\mathcal{C}} + (1-\alpha_t) P_{\mathcal{C}},$$

so the top eigenvalue satisfies:

$$\lambda_{\max}(P_{\mathcal{C}}\Sigma_t P_{\mathcal{C}}) = \alpha_t \lambda_{\max}^{\mathcal{C}} + (1 - \alpha_t).$$

Hence,

$$c \leq 2(\alpha_t \lambda_{\max}^{\mathcal{C}} + (1 - \alpha_t)) \cdot \|G\|_F^2.$$

Substituting into our earlier expression for $\|\Delta W\|_F$, we obtain:

$$\|\Delta W\|_F \geq \frac{\|G\|_F^3}{2(\alpha_t \lambda_{\max}^{\mathcal{C}} + (1 - \alpha_t)) \cdot \|G\|_F^2} = \frac{\|G\|_F}{2(\alpha_t \lambda_{\max}^{\mathcal{C}} + (1 - \alpha_t))}.$$

This completes the proof. $\qquad\square$

| Method | Erase Object 3 | Erase Object 5 |
|--------|----------------|----------------|
| ESD | $0.0002 \pm 0.0003$ | $0.0000 \pm 0.0000$ |
| MACE | $0.0030 \pm 0.0031$ | $0.0040 \pm 0.0020$ |
| SDD | $0.0045 \pm 0.0026$ | $0.0077 \pm 0.0093$ |
| UCE | $0.0037 \pm 0.0008$ | $0.0003 \pm 0.0006$ |
| EraseDiff | $0.0000 \pm 0.0000$ | $0.0000 \pm 0.0000$ |
| SHS | $0.0000 \pm 0.0000$ | $0.0000 \pm 0.0000$ |
| SalUn | $0.0000 \pm 0.0000$ | $0.0000 \pm 0.0000$ |

Table 4: Incidental concept resurgence for Stable Diffusion v1.4. We compute the number of examples where the model after fine-tuning generated a concept that was unlearned when prompted with an unrelated concept (i.e. retained object) where the model before fine-tuning did not do this. This is over 2000 prompts.

## I  Incidental Concept Resurgence Results

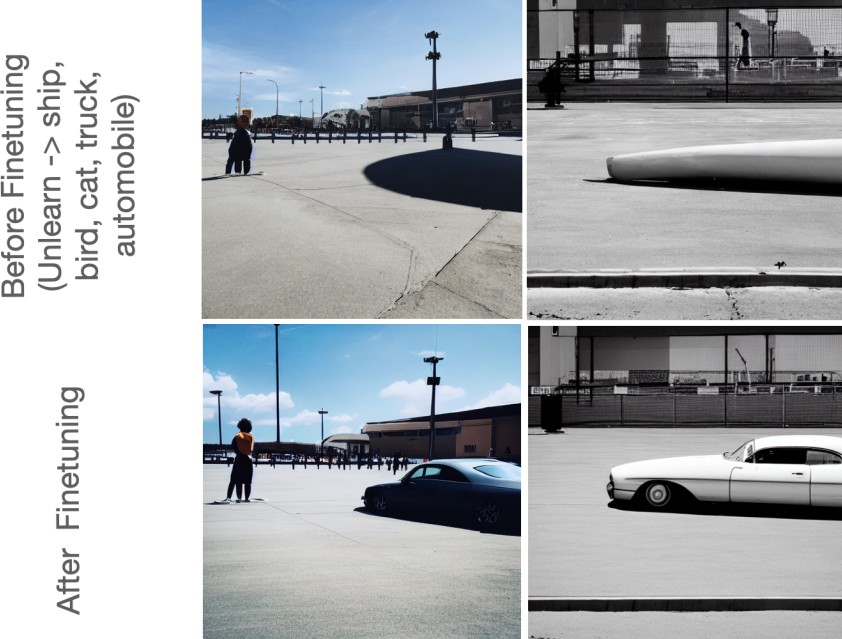

Figure 18: Unlearned concepts can resurge in benign prompts that don't contain the concept itself. Here we unlearned automobile using MACE on SDv1.4 and found that even on a benign prompt of "A photo of an airplane" the unlearned concept resurges after fine-tuning. Emphasizing the harm that resurgence can cause to downstream users.

## J   Additional Finetuning Hyperparameter Analyses

We demonstrate initial analyses on MACE for erasing 10 celebrities of varying fine-tuning parameters, showing consistent resurgence.

### J.1   Finetuning Dataset Size

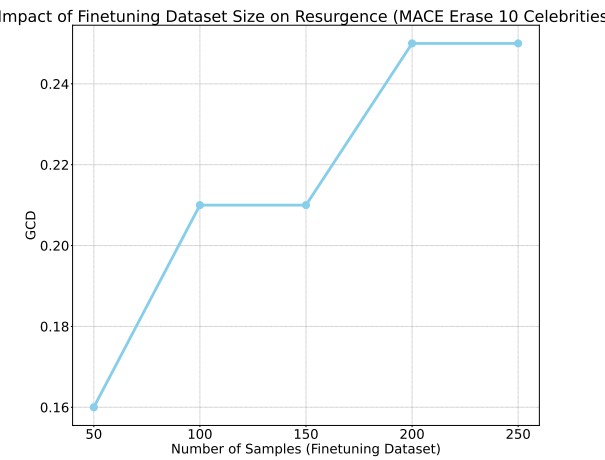

Figure 19: Increasing the number of samples results in more resurgence (higher GCD). Even with only 50 samples for 1000 fine-tuning LoRA steps there is 16% resurgence.

### J.2   Number of Finetuning Steps

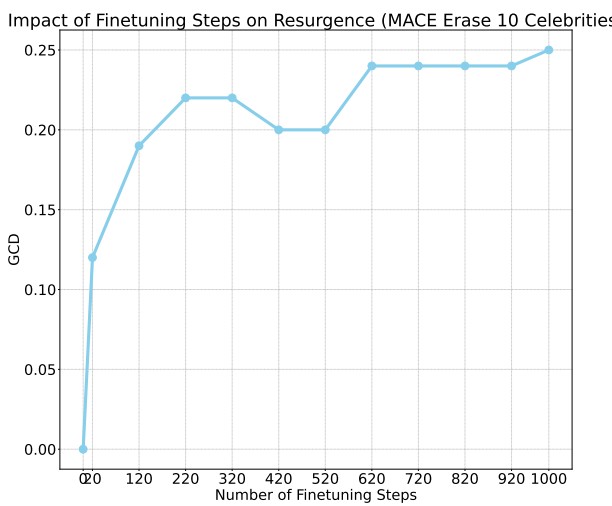

Figure 20: Increasing the number of steps results in more resurgence (higher GCD). Even with only 20 steps for 250 samples there is 12% resurgence.

# K   Additional Algorithm Results

| Algorithm | Erase Setting | Before FT | After FT |
|---|---|---|---|
| AdvUnlearn | Erase 1 | $0.074 \pm 0.000$ | $0.732 \pm 0.030$ |
| AdvUnlearn | Erase 5 | $0.292 \pm 0.036$ | $0.869 \pm 0.031$ |
| AdvUnlearn | Erase 10 | $0.376 \pm 0.031$ | $0.899 \pm 0.024$ |
| RGD | Erase 1 | $0.000 \pm 0.000$ | $0.000 \pm 0.000$ |
| RGD | Erase 5 | $0.081 \pm 0.067$ | $0.756 \pm 0.037$ |
| RGD | Erase 10 | $0.107 \pm 0.053$ | $0.810 \pm 0.033$ |

Table 5: Performance before and after benign fine-tuning for the erase 1, 5, 10 celebrity tasks for newer unlearning algorithms AdvUnlearn and RGD that are designed for robustness. Values are reported as mean $\pm$ standard deviation.

