# OpenReview forum: "Unstable Unlearning: The Hidden Risk of Concept Resurgence in Diffusion Models"
_TMLR — Accepted by TMLR_

### Review · Reviewer_aTUr · 2025-11-03

**Summary Of Contributions:**

- The paper shows a critical vulnerability in unlearning text-to-image diffusion models termed “Concept Revival.”
- An unlearned model, when finetuned, can cause it to relearn the unlearned concept. This is a critical problem, and the authors investigated factors such as scale and how the algorithmic choice impacts it.
- To explain why this phenomenon occurs, the authors attempted to offer theoretical intuition/proofs.

**Audience:**

No

**Audience Explanation:**

- Much of the information presented in this work is well-known within the Unlearning community.
- This work is missing multiple recent works that are very related. Already, some works have shown the resurgence phenomenon in Diffusion Models [2,3].
- Then, even in other modalities, there have been works which has shown this phenomenon [4,5].
- There is also another work [6], that shows after unlearning without any finetuning, there are latents present that can still produce the unlearned concept.
- This work doesn’t propose any methods to address or mitigate this phenomenon.
- Recent works have even tried to mitigate this resurgence phenomenon (e.g., [3]).

**Claims And Evidence:**

Yes

**Claims Explanation:**

- The authors have conducted a number of experiments using various methods to demonstrate this phenomenon across different unlearning methods and shown it successfully.
- However, I am not convinced about the reported Incidental Concept resurgence; the experimental setup isn’t very clear. It appears that sequential unlearning is performed on this set {cat, truck, automobile, ship, bird}. If this phenomenon had been observed in single
unlearning, it would have been more relevant, as existing literature has already shown that current unlearning methods aren’t very effective in sequential unlearning (check Table A5 in [1]).
- Additionally, this may also depend on the mapping concept during the unlearning process and the fine-tuning concept. Even the numbers shown in Table 4 are very low, as none of them are even close to 1%.
- Additionally, the existing literature and benchmarks currently use Classifiers [1].
- It would be helpful to provide some motivation for Using Molmo.
- Also, provide the numbers produced by the original SD Model on using your set of prompts and Molmo for evaluation.
- The authors have claimed in the introduction that concept resurgence happens even when finetuned on unrelated concepts, but in the supplementary material for finetuning, the authors have mentioned they used other celebrity concepts in case of celebrities, other prompts from i2p for Unsafe content erasure, and other fictional characters for copyright erasure. Calling this an unrelated concept is not entirely accurate because these concepts are closely related.

**References**
1. Zhang, Yihua, et al. &quot;Unlearncanvas: A stylized image dataset to benchmark machine unlearning for diffusion models.&quot; NeurIPS. 2024.
2. George, Naveen, et al. &quot;The Illusion of Unlearning: The Unstable Nature of Machine Unlearning in
Text-to-Image Diffusion Models.&quot; Proceedings of the Computer Vision and Pattern Recognition
Conference. 2025.
3. Gao, Hongcheng, et al. &quot;Meta-unlearning on diffusion models: Preventing relearning unlearned
concepts.&quot; Proceedings of the IEEE/CVF International Conference on Computer Vision. 2025.
4. Siddiqui, Shoaib Ahmed, et al. &quot;From Dormant to Deleted: Tamper-Resistant Unlearning Through
Weight-Space Regularization.&quot; arXiv preprint arXiv:2505.22310 (2025).
5. Deeb, Aghyad, and Fabien Roger. &quot;Do unlearning methods remove information from language
model weights?.&quot; arXiv preprint arXiv:2410.08827 (2024).
6. Rusanovsky, Matan, et al. &quot;Memories of forgotten concepts.&quot; Proceedings of the Computer Vision
and Pattern Recognition Conference. 2025.
7. Ko, Myeongseob, et al. &quot;Boosting alignment for post-unlearning text-to-image generative
models.&quot; Advances in Neural Information Processing Systems 37 (2024): 85131-85154.
8. Fan, Chongyu, et al. &quot;Towards llm unlearning resilient to relearning attacks: A sharpness-aware
minimization perspective and beyond.&quot; ICML (2025).
9. Zhang, Yimeng, et al. &quot;Defensive unlearning with adversarial training for robust concept erasure
in diffusion models.&quot; Advances in neural information processing systems 37 (2024).
10. Huang, Chi-Pin, et al. &quot;Receler: Reliable concept erasing of text-to-image diffusion models via
lightweight erasers.&quot; European Conference on Computer Vision. Cham: Springer Nature
Switzerland, 2024.
11. Tianqi Chen, et al. “Score forgetting distillation: A swift, data-free method for machine unlearning
in diffusion models.” In The Thirteenth International Conference on Learning Representations,
2025

**Requested Changes:**

- Please address the points/concerns raised in the above section(s) of this review.
- There are some recent works on adversarially robust unlearning [9,10] that have not been explored in this work. It would be interesting if such a phenomenon occurs in these works as well.
- SFD [11] is another paper that performs unlearning very well and is adversarial robust, so it would be great to know if such a phenomenon occurs in this case, as the algorithm differs significantly from existing ones.
- Section 5 is insightful. Related work pursuing a similar perspective includes [7,4,8].
- The authors can evaluate whether the reported Concept resurgence persists in [7], as this targets diffusion models, a direct empirical test should be feasible.

**Minor Changes/edits**
1) Repetition of Citation 31 can be corrected.

---

### Review · Reviewer_9e8S · 2025-11-06

**Summary Of Contributions:**

This paper identifies an important and interesting issue: while current unlearning approaches appear effective at removing certain concepts, they often fail to completely prevent resurgence—the reappearance of forgotten concepts.

**Additional Comments:**

None.

**Audience:**

Yes

**Audience Explanation:**

Unlearning is a critical and timely topic for image generation models. This study raises awareness of fundamental weaknesses in current unlearning techniques, which could inspire the research community to develop more robust and reliable solutions.

**Broader Impact Concerns:**

None.

**Claims And Evidence:**

Yes

**Claims Explanation:**

The paper conducts a thorough empirical analysis of multiple unlearning algorithms across several key scenarios, including Celebrity, Copyright, Object, and Unsafe content. The experiments convincingly demonstrate that most existing methods suffer from the resurgence issue, thereby supporting the authors’ claims.

**Requested Changes:**

While the findings are important and insightful, the paper’s quality could be further improved through additional analysis and clearer presentation. My specific suggestions are as follows:
* Parameter analysis: It would be valuable to explore whether certain parameter configurations can prevent fine-tuning–based approaches from exhibiting the resurgence issue. This discussion is currently missing.
* Figure quality and clarity: Figures such as 16 and 17 are of poor visual quality and have low resolution, which significantly reduces readability. Additionally, the legend shows both blue and orange bars, but only orange bars appear in the figure. The figure captions are also vague—for instance, the statement “unlearning all on all the parameters helps further” should be clarified.
* Contradiction in Section 5: The paper claims that if the projection of S onto C is non-zero, resurgence occurs. However, Table 3 shows that fine-tuning UCE with all parameters does not necessarily result in noticeable resurgence. This observation contradicts the stated proposition, limiting its explanatory power for nonlinear models used in practice.
* Comparison between fine-tuning and closed-form methods: Additional analysis contrasting these two paradigms would strengthen the paper. For example, why do closed-form methods remain robust even when followed by fine-tuning, as observed in Table 3?
* High-level takeaway in Section 6: While Section 6 primarily discusses limitations, it would be helpful to include a concise summary of key takeaways. For instance, should future research focus on closed-form approaches due to their robustness, or are there promising directions to mitigate resurgence in fine-tuning–based methods?

---

### Review · Reviewer_KkG8 · 2025-11-10

**Summary Of Contributions:**

This paper introduces `concept resurgence', a critical and previously overlooked vulnerability in text-to-image diffusion models. The authors demonstrate that concepts supposedly ``unlearned'' from a model can be unintentionally reintroduced through subsequent, benign fine-tuning on unrelated data.

The paper's primary strength, and its most distinct contribution, lies in its theoretical analysis in Section 5. The proposed mathematical model provides a fundamental explanation for why this phenomenon occurs. The authors use a linear score-based model to provably show that resurgence is driven by two key factors:

- Gradient Resurgence Bound: The analysis identifies that any non-zero overlap ($\gamma(\mathcal{S},\mathcal{C}) > 0$) between the fine-tuning gradient subspace ($\mathcal{S}$) and the forgotten concept's subspace ($\mathcal{C}$) is sufficient to reintroduce gradients associated with the forgotten concept  This makes resurgence an inevitable consequence of high-dimensional subspace alignment, not just an anomaly.

- Curvature-Limited Sensitivity Bound: This bound provides the crucial ``amplification'' mechanism. It demonstrates that even small, stray gradients in the forgotten subspace can induce disproportionately large parameter updates if the loss landscape in that direction has low curvature (i.e., is flat)

**Audience:**

Yes

**Audience Explanation:**

Those working on concept forgetting would find this work very relevant.

**Broader Impact Concerns:**

There is no explicit Broader Impact and Concerns section. But the authors discuss the relevant points in the second paragraph of their Discussion and Limitation Section.

**Claims And Evidence:**

Yes

**Claims Explanation:**

The claims for ``concept resurgence'' are well-supported by both quantitative results, such as those in Table 1, and qualitative examples. The theoretical claims in Section 5, which explain \textit{why} resurgence occurs, are explicitly proved for a linear score-based diffusion model; the authors use this simplified setting to derive their formal bounds.

**Requested Changes:**

There is no doubt that the contribution of this paper—exploring "Concept Resurgence"—is important and original. However, this phenomenon has been explored in a contemporaneous paper published in CVPR 2025 [1]. Please cite the following work and revise your introduction and related work sections to properly contextualize your contribution in light of this simultaneous discovery. Acknowledging this work will help clarify your paper's unique contribution.


[1] N. George, K.N. Dasaraju, R.R. Chittepu, and K.R. Mopuri, "The Illusion of Unlearning: The Unstable Nature of Machine Unlearning in Text-to-Image Diffusion Models," in Proceedings of the Computer Vision and Pattern Recognition Conference (CVPR), 2025, pp. 13393–13402.